# Metal-ligand dual-site single-atom nanozyme mimicking urate oxidase with high substrates specificity

Kaiyuan Wang[1], Qing Hong[1], Caixia Zhu[1], Yuan Xu[1], Wang Li[1], Ying Wang[1], Wenhao Chen[1], Xiang Gu[1], Xinghua Chen[1], Yanfeng Fang[1], Yanfei Shen [2] ✉, Songqin Liu[1] & Yuanjian Zhang [1,3] ✉

In nature, coenzyme-independent oxidases have evolved in selective catalysis using isolated substrate-binding pockets. Single-atom nanozymes (SAzymes), an emerging type of non-protein artificial enzymes, are promising to simulate enzyme active centers, but owing to the lack of recognition sites, realizing substrate specificity is a formidable task. Here we report a metal-ligand dual-site SAzyme (Ni-DAB) that exhibited selectivity in uric acid (UA) oxidation. Ni-DAB mimics the dual-site catalytic mechanism of urate oxidase, in which the Ni metal center and the C atom in the ligand serve as the specific UA and $O_2$ binding sites, respectively, characterized by synchrotron soft X-ray absorption spectroscopy, in situ near ambient pressure X-ray photoelectron spectroscopy, and isotope labeling. The theoretical calculations reveal the high catalytic specificity is derived from not only the delicate interaction between UA and the Ni center but also the complementary oxygen reduction at the beta C site in the ligand. As a potential application, a Ni-DAB-based biofuel cell using human urine is constructed. This work unlocks an approach of enzyme-like isolated dual sites in boosting the selectivity of non-protein artificial enzymes.

Artificial enzymes, which are catalysts utilizing nonenzymatic proteins or non-protein materials to mimic natural enzymes, have been recognized as potential substitutes for natural enzymes[1–11]. Protein-based artificial enzymes still suffer from being costly to store, unstable, and sensitive to harsh physiochemical conditions. Therefore, non-protein artificial enzymes, such as emerging nanozymes[12–20], clusterzymes[3,21,22], and single-atom nanozymes (SAzymes)[23–29] are attracting increasing interest. Among them, SAzymes, which feature mononuclear metal sites and well-defined coordination structures and are skilled in simulating enzyme active centers[23,27,28,30–32]. This makes the SAzymes activity approaching or even surpassing that of natural enzymes in industrial reactions[30,33,34]. However, unlike natural enzymes, most non-protein artificial enzymes lack any recognition sites for specific substrates. This limitation prevents them from

specific catalysis and diminish atom economy in industrial applications[13,35,36].

To address this critical issue, several seminal studies, including importing extrinsic molecular recognition units (e.g., molecular imprinting[37,38] and macromolecular[39])[40–42] and the engineering of intrinsic active sites (e.g., coordination environments and dopant regulation)[43–45] have been reported. Despite great success, these methods are either impeded by limited types of artificial recognition units or merely engineerable to a given type of catalyst. Therefore, the development of non-protein artificial enzymes with high specificity remains challenging[32].

Mimicking the catalytic mechanisms of natural enzymes can unlock new possibilities. Generally, enzymatic reactions are extremely specific and efficient. Oxidases, for example, are a dual-substrate

[1]Jiangsu Engineering Research Center for Carbon-Rich Materials and Devices, Jiangsu Province Hi-Tech Key Laboratory for Bio-Medical Research, School of Chemistry and Chemical Engineering, Nanjing 211189, China. [2]Medical School, Southeast University, Nanjing 210009, China. [3]Department of Oncology, Zhongda Hospital, Southeast University, Nanjing 210009, China. ✉e-mail: Yanfei.Shen@seu.edu.cn; Yuanjian.Zhang@seu.edu.cn

enzyme that relies on oxygen ($O_2$) to catalyze the oxidation of small molecule substrates. Among them, urate oxidase (UOX), which plays a vital role in the metabolism of uric acid (UA) without cofactors, catalyzes the oxidation of UA with the concomitant reduction of $O_2$ to $H_2O_2$[46–48]. UOX has a UA-binding site consisting of Arg-176, Gln-228, and Val-227 residues, and an $O_2$-binding site consisting of Thr-57*, Asn-254, and His-256 residues[49]. These two sites are isolated and selective, allowing UOX to only catalyze the oxidation of UA with $O_2$[50]. In contrast, such a dual-site mechanism has been seldom explored for oxidase-like mimics. Even under the very recent single-atom catalysts (SACs) consisting of metal-metal dual sites, only a single substrate (usually $O_2$) was directly activated[26,37,51–55]. Since nonmetallic active sites are commonly existed in natural enzymes, one viable biomimicking strategy is to introduce a competitive nonmetallic site into SACs[56,57], and the primary prerequisite is to choose a catalyst with an unambiguous structure. However, traditional metal-nitrogen-doped carbon (M-N-C) SACs are prepared by high-temperature pyrolysis with randomly dispersed metal atoms and no clear continuous local structures. This prevents them from being precisely engineerable. Old polymeric catalysts that have precise engineerable molecular structures come to our attention[58]. Among them, coordination polymers (CPs) with uniform distribution, high density, unambiguous single-atom framework of metal centers and ligands, and ligand-centered redox activity would be a class of ideal candidates for studying the nonmetallic site of SACs[59–64]. Many catalysts based on coordination polymers have been reported; nonetheless, the studies on enhancing the specificity of enzyme-like reaction based on polymeric catalysts are rare.

Herein, we report a coordination polymer-based dual-site SAzyme consisting of Ni metal centers and 3,3′-diaminobenzidine ligands (Ni-DAB) for specific UA oxidation. Comprehensive experiments and theoretical calculations unambiguously unveiled that the Ni metal centers and the beta C atoms in the ligand serve as the specific UA and $O_2$ binding sites, respectively. Such metal-ligand dual sites successfully endowed Ni-DAB with UOX-like high catalytic specificity for UA oxidation (Fig. 1a). As a proof-of-the-concept application, a biofuel cell using human urine was constructed, which successfully drove a temperature and humidity sensor. This work would open a avenue to boost the specificity of non-protein artificial enzymes by mimicking the catalytic mechanism of natural enzymes.

## Results

### Synthesis and structural characterizations of Ni-DAB

Briefly, Ni-DAB were fabricated using nickel (II) salts and 3,3′-diaminobenzidine (DAB) in the presence of ammonia and stirred for 3 h with air bubbling (Fig. 1b). The obtained black powdered product was named Ni-DAB. As controls, Co-DAB and Ni-BTA (1,2,4,5-benzenetetramine) were prepared using the same method by changing the central metal or ligand, respectively. Fourier-transform infrared spectroscopy (FTIR) and nuclear magnetic resonance spectroscopy (NMR) were used to confirm the coordination interactions between Ni and DAB. In the FTIR spectra (Supplementary Fig. 1), the two characteristic N-H stretching modes from $-NH_2$ disappeared in Ni-DAB, whereas other phenyl-related vibrations remained. In the $^1H$ NMR spectra (Fig. 1c), the typical signals for the aromatic H in DAB at 6.4-6.8 ppm gradually broadened and downshifted when $Ni^{2+}$ was added to the DAB solution, and the $-NH_2$ proton signal of DAB at 4.4 ppm quickly disappeared, which was attributed to the rapid H/D exchange between $-NH_2$ and $D_2O$. These results confirmed the coordination interaction between Ni and DAB in Ni-DAB.

Scanning electron microscopy (SEM, Supplementary Fig. 2) and high-resolution transmission electron microscopy (HR-TEM, Supplementary Fig. 3) images showed that Ni-DAB appeared as conglomerated particles with an average size of 200-300 nm. The corresponding TEM elemental mapping images demonstrated that the

C, N, Ni, and Cl species were homogeneously dispersed across the entire Ni-DAB particle (Supplementary Fig. 3b). The morphology of the control samples Co-DAB and Ni-BTA were illustrated in Supplementary Fig. 4, both of which were composed of irregular particles. The large-area high-angle annular dark-field scanning transmission electron microscopy (HAADF-STEM) image confirmed the existence of uniformly dispersed single-atom Ni on Ni-DAB. The individual bright dots (highlighted by yellow circles) illustrated the atomic dispersion of single Ni atoms in Ni-DAB (Fig. 1d).

The X-ray photoelectron spectroscopy (XPS) provided electronic structure information on Ni-DAB. The Ni 2$p$ spectrum of Ni-DAB (Supplementary Fig. 5) exhibited Ni 2$p^{1/2}$ (873.1 eV) and 2$p^{3/2}$ (855.9 eV) peaks, indicating the presence of $Ni^{2+}$. The N 1$s$ spectrum could be deconvoluted into three peaks, the peak at 398.5 eV can be attributed to the coordination between $-NH_2$ and Ni in the Ni-$N_x$ motif. The other two peaks at 399.3 eV and 400.6 eV were attributed to C = N and C-N, respectively (Supplementary Fig. 6a and Note 1). XPS survey in Supplementary Fig. 7 suggested that Ni-DAB contains charge-balancing anions ($Cl^-$), differing from those of the control samples Co-DAB and Ni-BTA, which was reaffirmed by energy-dispersive X-ray spectroscopy (EDS, Supplementary Fig. 8). This implied that Ni-DAB was positively charged. Furthermore, after UA was catalyzed by Ni-DAB in alkaline conditions, $Cl^-$ in Ni-DAB was almost completely replaced by $OH^-$, suggesting that $Cl^-$ only had an equilibrium charge effect, did not coordinate to the metal center of Ni-DAB in an axial adsorption manner[65] (Supplementary Fig. 9, Note 2 and Table 1), and had no effect on the catalytic performance. The zeta potential of Ni-DAB was measured to be +30 mV, whereas Co-DAB and Ni-BTA only had low negative potentials (Supplementary Fig. 10). In general, Ni-$N_x$-based CPs usually exhibit a strong electron paramagnetic resonance spectroscopy (EPR) signal originating from a single electron of the ligand[59,61]. As shown in Fig. 1e, compared with the typical CPs Ni-BTA, a weak EPR signal was observed for Ni-DAB, indicating that the Ni-DAB ligand lost its single electron. These results confirmed that the Ni-DAB ligand was in an electron-deficient oxidation state.

The valence state and coordination structure of the metal center in Ni-DAB were identified using synchrotron X-ray absorption spectroscopy (XAS). As shown in Fig. 1f, the absorption edge of Ni-DAB in the X-ray absorption near-edge structure (XANES) was located between the control samples of Ni foil and $Ni_2O_3$, which was close to that of the Ni-phthalocyanine complex (NiPc), declaring that the Ni valence state was situated between $Ni^0$ and $Ni^{3+}$. More quantitative structural information on Ni coordination was obtained using Fourier-transformed extended X-ray absorption fine structure (EXAFS) (Fig. 1g). The major peak (-1.41 Å) and the secondary peak (-2.25Å) were attributed to the backscattering of light atoms (N and C) situated in first and second coordination shell of the Ni. The Ni-Ni peak at ca. 2.15 Å and the Ni-O peak at ca. 1.65 Å were absent in Ni-DAB, demonstrating that the Ni atoms were atomically dispersed in Ni-DAB. This result echoed the observations from HAADF-STEM analyses. The chelation of amino groups made Ni in Ni-DAB present a unimodal in the first shell like the pyrolysis-prepared Ni-N-C SACs[66], enabling Ni-DAB to only have the Ni-N coordination. The Ni-$N_x$-$C_x$ model (Supplementary Fig. 11) was then used to fit the metal K-edge EXAFS spectra (Fig. 1h). The result showed that the model fitted well with the experimental data. In the first shell, the coordination number of Ni was 4.0 and the average bond length of Ni-$N_x$ was 1.84 Å, aligning seamlessly with the quadrilateral configuration. Moreover, a Ni···C spacing of 2.68 Å was also deduced in the second shell, corresponding to a further Ni···C distance. These results revealed that Ni metal center in Ni-DAB was involved in the Ni-$N_4$-$C_4$ coordination structure (Supplementary Table 2). The stable configuration of Ni-DAB was further verified by the first principle of molecular dynamics, consistent with EXAFS fitting (Supplementary Fig 12 and Note 3).

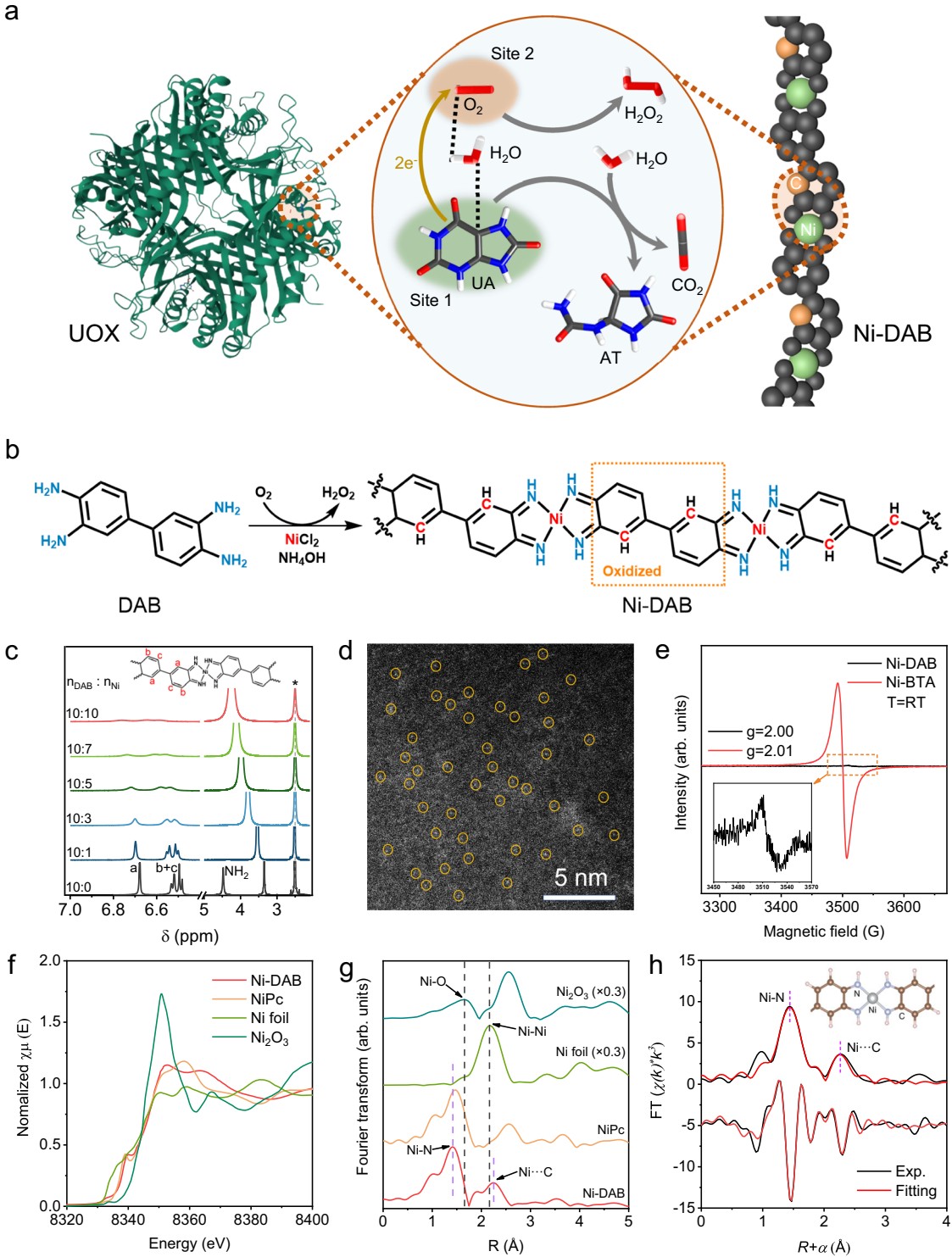

**Fig. 1 | Synthesis and characterization of Ni-DAB. a** Scheme of catalytic UA oxidation by the isolated dual sites of natural UOX and Ni-DAB. For clarity, catalysts and reactants were not drawn in scale, and except for reactive C (yellow) and Ni (green) sites, all other atoms in Ni-DAB are in gray. For substrates and products, the white, grey, blue, and red sticks represent H, C, N, and O atoms, respectively. Image of UOX from the RCSB PDB (RCSB.org) of PDB ID 2ZKA (Colloc'h, N., L. Gabison et al. (2008) Biophys J 95: 2415-2422). **b** Synthetic scheme of Ni-DAB. **c** $^1$H NMR spectra of polymeric Ni-DAB complex with different DAB/Ni molar ratios from 10:0 to 10:10. **d** HAADF-STEM image of Ni-DAB. The experiment was repeated 3 times independently with similar results. **e** EPR spectra of Ni-DAB and Ni-BTA acquired at room temperature. **f** Ni K-edge XANES spectra of Ni-DAB, NiPc, Ni foil and $Ni_2O_3$. **g** FT-EXAFS spectra for the Ni K-edge of Ni-DAB. **h** First-shell (Ni−N) and second-shell (Ni···C) fitting of FT-EXAFS spectra for Ni-DAB. "arb. units" refers to arbitrary units. Source data are provided with the paper.

According to elemental analyses (Supplementary Table 3), the quantitative atomic ratio of Ni, N, and C was close to 1:5:15 (Theoretical results with infinite polymerization degree was 1:4:12). Based on these results, Ni-DAB is a short chain composed of ca. five DAB ligands and four Ni atoms. Supplementary Fig. 13 shows a schematic diagram of the Ni-DAB chain, and the simulated $^{13}$C NMR spectrum of this chain was well-fitted to the solid $^{13}$C NMR spectrum of Ni-DAB, confirming the structure of this chain. Therefore, these comprehensive

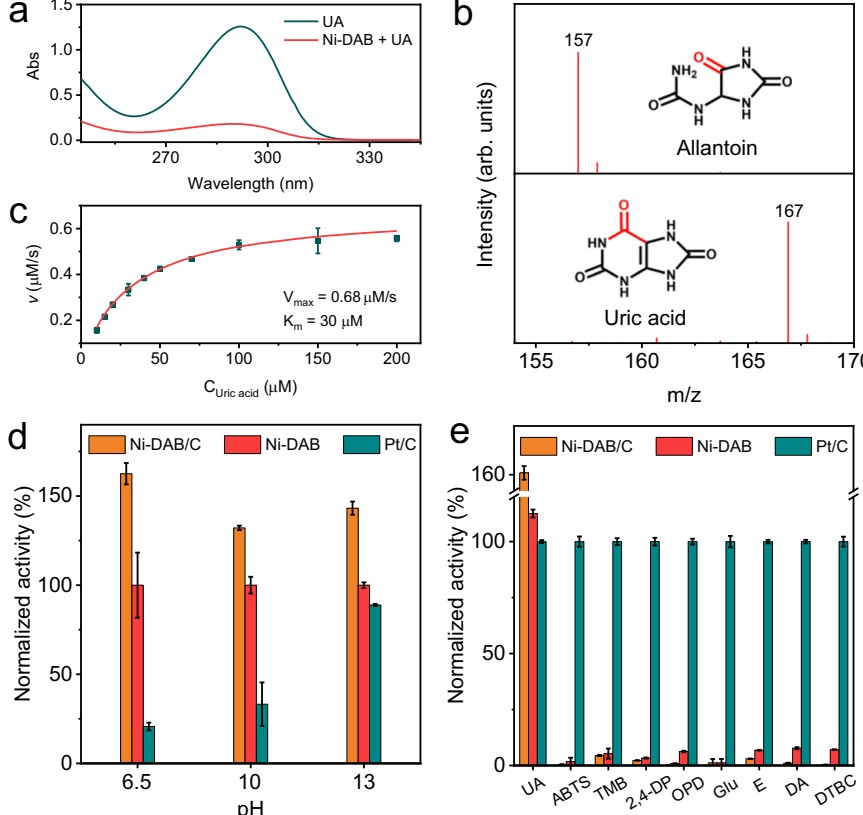

**Fig. 2 | UOX-like activity and specificity of Ni-DAB. a** Changes in UV-vis absorption spectra of UA catalyzed by Ni-DAB. **b** Mass spectra of UA and catalytic product of UA by Ni-DAB. **c** The Michaelis–Menten curves for UA catalysis by Ni-DAB. The solution was saturated with air (dissolved O$_2$: 8.5 mg/L) ($n = 3$ independent experiments). **d** Normalized UOX-like activity of Ni-DAB, Ni-DAB/C, and Pt/C at different pH value ($n = 3$ independent experiments). **e** The catalytic activity of various oxidase substrates by Ni-DAB, Ni-DAB/C, and Pt/C ($n = 3$ independent experiments). All data are presented as mean ± SD. "arb. units" refers to arbitrary units. Source data are provided with the paper.

characterizations collectively affirmed the formation of the proposed Ni-N$_4$-based CPs (Ni-DAB).

## UOX-like activity

The UOX-like activity of the Ni-DAB SAzyme was evaluated by the catalytic oxidation of UA. The UA solution exhibited a typical absorption peak at 293 nm, and the peak decreased significantly when it was oxidized to allantoin (AT, Fig. 2a and Supplementary Fig. 14) by Ni-DAB. Electrospray ionization mass spectrometry (ESI-MS) was used to validate the oxidation product. Before catalysis, a fragment of UA ([M-H]) was observed at 167. As expected, the oxidation product AT ([M-H]) was observed at 157 after catalysis using Ni-DAB (Fig. 2b). This observation matched the behavior of natural UOX in terms of oxidation product (Supplementary Fig. 15). The kinetic studies further demonstrated that Ni-DAB well followed the typical Michaelis-Menten kinetics model (Fig. 2c and Supplementary Fig. 16). Ni-DAB catalyzed UA oxidation with dependence on oxygen concentration like natural oxidase. In the air-saturated solution, the calculated value for $V_{max}$ was 0.68 μM s$^{-1}$. The UOX-mimic activity of Ni-DAB was significantly higher than those controls of Co-DAB, Ni-BTA (Supplementary Fig. 17), and previously reported artificial UOX mimics (Supplementary Table. 4). The specific activity value of Ni-DAB was determined to be 1.51 U mg$^{-1}$ (Supplementary Fig. 18). Notably, Pt-based catalysts have also been reported to exhibit an efficient UOX-like activity[67-69], then the commercial Pt/C (20%) was also used as a control. As shown in Fig. 2d and Supplementary Fig. 19, the UOX-like activity of Pt/C was worse than that of Ni-DAB at different pH values. The measured

specific activity of Ni-DAB was ca. 3-fold higher than that of the Pt/C (0.49 U mg$^{-1}$) (Supplementary Fig. 20). These results proved that Ni-DAB had a high UOX-like activity. Moreover, Ni-DAB remained high activity over a broad pH value (6 ~ 14) and temperature (10 ~ 80 °C) range, when the pH value was below 6, Ni-DAB became unstable and inactive (Supplementary Fig. 21). The UOX-like activity of Ni-DAB was stable and recyclable (Supplementary Fig. 22). The XPS analysis disclosed that the Ni 2$p$, C 1$s$ and N 1$s$ spectra of the spent Ni-DAB show no obvious changes compared with those of the fresh Ni-DAB (Supplementary Fig. 23). These results reconfirmed its robust structural stability during the catalytic UA oxidation. Besides, we found that the catalytic activity of Ni-DAB could be optimized by the degree of polymerization (Supplementary Note 4), and Ni-DAB with the highest activity was explored in this study unless otherwise specified (Supplementary Fig. 24 and 25).

## Catalytic specificity

To investigate the catalytic specificity property of Ni-DAB, a series of oxidase substrates including 2,2'-azino-bis(3-ethylbenzothiazoline-6-sulfonic acid) (ABTS), 3,3',5,5'-tetramethylbenzidine (TMB), 2,4-dichlorophenol (2,4-DP), o-phenylenediamine (OPD), glucose (Glu), epinephrine (E), dopamine (DA), and 3,5-ditert-butyl catechol (DTBC) were explored. As shown in Fig. 2e and Supplementary Fig. 26-33, compared to the high oxidase activity of non-selective Pt/C for all substrates, Ni-DAB showed surprisingly low activity toward these substrates except for UA. These results unveiled that Ni-DAB had good catalytic specificity for UA.

Nonetheless, a minor catalytic oxidation for other oxidase substrates was still observed for Ni-DAB. Control experiments disclosed that the DAB ligand was oxidized by dissolved $O_2$ during the synthesis of Ni-DAB. In Supplementary Fig. 34, when HRP and TMB were added to the supernatant reaction solution (RS) for the synthesis of Ni-DAB. An obvious characteristic adsorption peak of oxidized TMB ($TMB_{ox}$) appeared at 652 nm, implying that $H_2O_2$ was generated in the solution. This explained why the Ni-DAB ligand was in an electron-deficient oxidation state, because electrons from the ligand were transferred to $O_2$ during the synthesis of Ni-DAB. Furthermore, trace amounts of $H_2O_2$ were detected in the aqueous dispersion supernatant (SN) of as-synthesized Ni-DAB. And, the amount of $H_2O_2$ increased under $O_2$-saturated conditions, while decreased under $N_2$-saturated conditions (Supplementary Fig. 35a). EPR trapping measurements were further performed to detect free oxygen radicals in solution containing Ni-DAB. Supplementary Fig. 35b showed a signal typical of the DMPO-OOH adduct was detected in Ni-DAB dispersion. These results confirmed the generation of reactive oxygen species (ROS) from $O_2$ reduction by the as-synthesized Ni-DAB. We speculated that not all ligands in Ni-DAB were oxidized during synthesis and some of them continued to reduce $O_2$ in as-synthesized Ni-DAB, which generated ROS and resulted in an imperfect catalytic specificity in substrates. Therefore, to further improve the catalytic specificity, carbon black was added during the synthesis of Ni-DAB (Ni-DAB/C, Supplementary Fig. 36 and 37), which can increase the contact area between the DAB ligands and $O_2$ during the preparation process, making the Ni-DAB ligand oxidation be more complete. As a result, the specific surface area of Ni-DAB increased from 29.35 to 196.01 $m^2\ g^{-1}$ (Supplementary Fig. 38) and inhibited the non-selective catalytic activity of Ni-DAB to other substrates (Fig. 2e and Supplementary Fig. 25-32). Notably, the catalytic efficiency of UA oxidation was improved by 50% (Fig. 2d, Supplementary Fig. 19 and 39).

## Identification of metal-ligand dual active sites and catalytic mechanism

The addition of negatively charged substrates (UA, ABTS, etc) can cause zeta potential changes of Ni-DAB (Supplementary Fig. 40), but only UA can be catalyzed, indicating the selectivity was not caused by electrostatic adsorption. For the catalytic process by natural UOX, UA and $O_2$ join successively at respective binding site first, and UA transfers two electrons to $O_2$ through UOX. Then, a hydroxyl group derived from water is inserted at the 5-C position of UA. Finally, UA is converted to 5-hydroxyisourate (5-HIU) that then hydrolyzed into AT and $O_2$ is reduced to $H_2O_2$ (Fig. 3a)[46,48,50,70,71]. Owing to the specificity of Ni-DAB, we conjectured that Ni-DAB with the isolated dual sites has a comparable catalytic mechanism to natural UOX.

Figure 3b disclosed that EPR signal of Ni-DAB was enhanced by the addition of UA to Ni-DAB under deoxygenation conditions. This enhanced signal came from the electron transfer from UA to the DAB ligand. Further, when UA was added to the Ni-DAB dispersion, more $H_2O_2$ and superoxide radical ($\cdot O_2^-$) were detected than in the aqueous dispersion supernatant of as-synthesized Ni-DAB (Supplementary Fig. 34 and 41). These results implied that Ni-DAB has an UOX-like electron transfer mechanism. In addition, an isotope labeling experiment was performed to verify whether Ni-DAB catalyzed UA oxidation by inserting a water-derived hydroxyl group at the 5-C position like natural UOX. It was assumed when $H_2^{18}O$ was used, AT marked by $^{18}O$ (AT-$^{18}O$) would be produced if this was the case. In Fig. 3c, the fragment of the oxidation product AT-$^{18}O$ ([M − H]) was observed at m/z = 159 by ESI-MS, suggesting the inserted hydroxyl group in UA was from water, in coincidence with the natural UOX. These results strengthened our conjecture that Ni-DAB has an UOX-like catalytic mechanism in view of the electron transfer and origin of the hydroxyl group.

Next, poison experiments and electrochemical oxygen reduction reactions (ORR) were carried out to validate the isolated dual

sites in Ni-DAB. The thiocyanate ions ($SCN^-$) as an inhibitor were supposed to block the Ni metal centers[72,73]. Figure 3d showed the catalytic oxidation activity of Ni-DAB was inhibited after adding $SCN^-$ in the reaction solution. Control experiments using natural UOX and UA showed that $SCN^-$ did not impair the substrates and products (Supplementary Fig. 42). This revealed that the metal centers have a key role in the UOX-like activity of Ni-DAB. Subsequently, ORR experiments showed that unlike Pt/C (Supplementary Fig. 43), $SCN^-$ did not impair the ORR performance of Ni-DAB/C in both alkaline and neutral conditions (Supplementary Fig. 44), confirming the weak role of the metal centers in the ORR. This indicated that rather than the metallic sites, the ORR active centers for Ni-DAB were the DAB ligands. Therefore, the UOX-like activity of Ni-DAB utilized the metal-ligand dual sites, where UA bound at the metal center and $O_2$ bound at the ligand.

To further determine the precise active site for $O_2$ adsorption in the DAB ligand, $O_2$ temperature-programmed desorption mass spectrometry ($O_2$-TPD-MS), soft O K-edge XANES, and in situ near ambient pressure X-ray photoelectron spectroscopy (NAP-XPS) were explored. Supplementary Fig. 45a showed a MS fragment peak with m/z of 32 in the range of 60 ‐ 240 °C, which was attributed to adsorbed oxygen. In thermal gravimetric analyzer (TGA) curves, the weight loss of Ni-DAB under 150 °C could be ascribed to adsorbed gases, while it was not observed in DAB (Supplementary Fig. 45b and Note 5). Meanwhile, the TGA cycle tests at 30 ‐ 150 °C indicated that Ni-DAB can adsorb oxygen reversibly (Supplementary Fig. 45c). The soft O K-edge XANES provided information about the species with oxygen binding to Ni-DAB. As shown in Fig. 3e, the clear O signal confirmed the existence of oxygen-containing groups and oxygen species. The sharp peak at about 531.8 eV was assigned to the π* excitation of C = O or −COOH. The peak at about 534.5 eV was attributed to π* excitation in charge transfer between C and O in C = O or C-O bonds. The broad peak at about 539.6 eV corresponded to the σ* excitation of C − O[57,74−76]. For in situ NAP-XPS, the initial XPS spectra of Ni-DAB were firstly measured. Subsequently, the SAzyme was in situ thermally treated at 150 °C for 2h in vacuum to remove adsorbed oxygen species. After thermal treatment, the intensity of the O 1s spectrum (Supplementary Fig. 46a) dramatically decreased. The C 1s spectrum intensity decreased at the high-binding-energy side and increased at the low-binding-energy side, and the area of integration remained constant (i.e., 1445:1447, Fig. 3f). This indicated that the amounts of C atoms remained constant and the chemical state changed. The differential analysis showed the intensity of C-O and C = O signal decreased, and that of C-N signal increased (Fig. 3f), elucidating the bonded oxygen atoms on C were removed by the temperature increase. Afterward, the treated sample was exposed to 0.5 mbar $O_2$ gas for 30 min, the O 1s intensity was slightly recovered (Supplementary Fig. 46b), and the C 1s spectrum had a slight positive shift (Supplementary Fig. 47). As a control, in all stages of testing, no noticeable changes in the Ni 2p and N 1s spectra were noted (Supplementary Fig. 48). Therefore, the binding site for O species of Ni-DAB was on the C site in the DAB ligand.

In organisms, some oxidases apply coenzymes as electron acceptors instead of oxygen. To validate the specificity of the C site for $O_2$, the coenzymes as electron acceptors were explored, including nicotinamide adenine dinucleotide ($NAD^+$) and cytochrome c (Cyt c). Supplementary Fig. 49 exhibited that $NAD^+$ and Cyt c with typical absorption peaks at 258 nm and 405 nm, respectively, which disappeared when they were reduced. When $NAD^+$ and Cyt c were added to the deoxygenation catalytic system of UA by Ni-DAB, no significant changes of their concentrations were observed in 30 min (Fig. 4g). In contrast, the concentration of $O_2$ gradually decreased over time. These results indicated that coenzymes cannot replace $O_2$ as electron acceptors for Ni-DAB. This implied the C site of Ni-DAB had a high specificity for $O_2$, similar to that in the natural UOX.

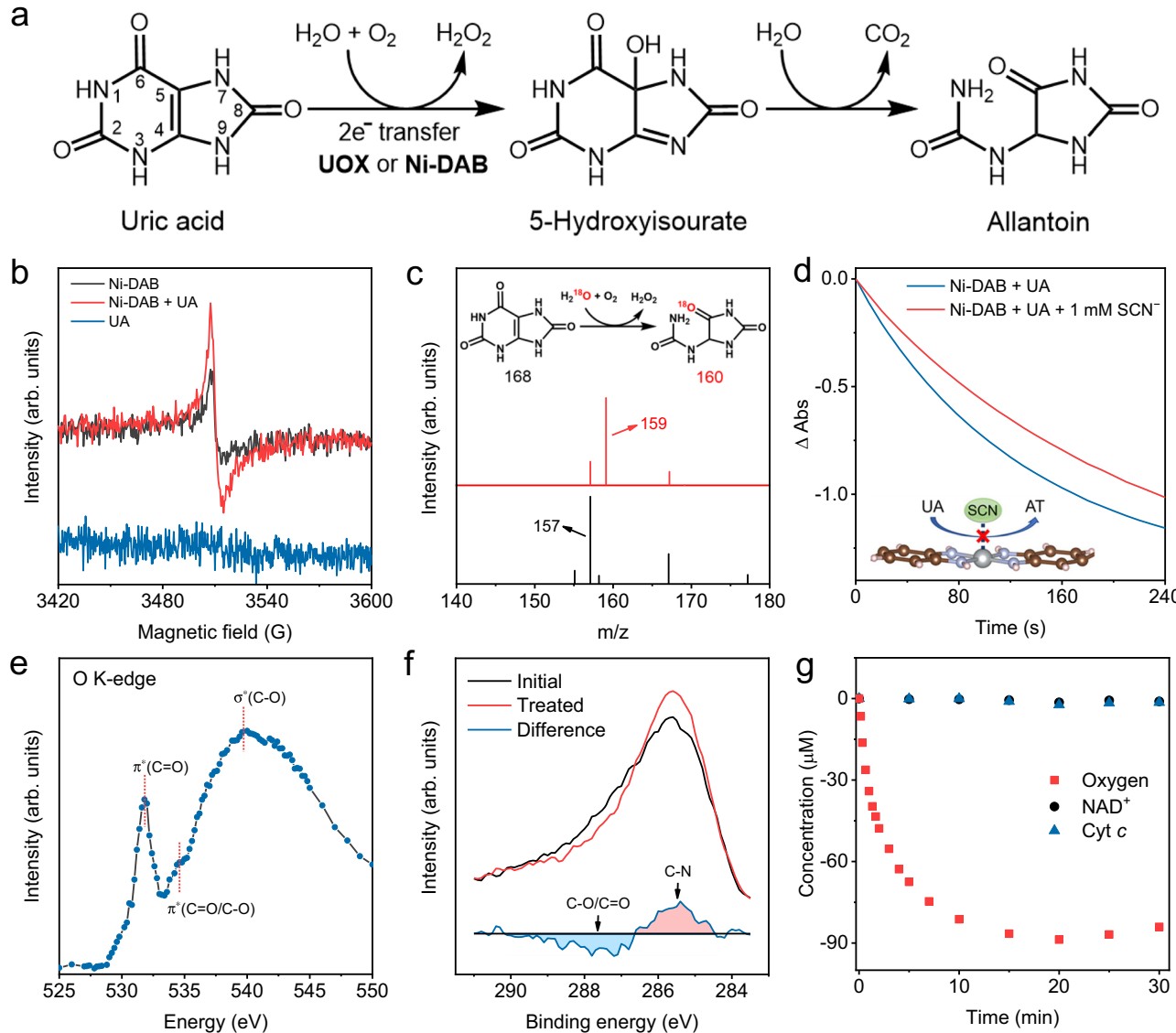

**Fig. 3 | Catalytic mechanism of Ni-DAB. a** Equation for natural UOX catalyzed UA oxidation process. **b** EPR spectra of Ni-DAB with UA added under deoxygenation conditions. **c** Mass spectra of catalytic products of UA by Ni-DAB in $H_2O$ (black curve) or $H_2^{18}O$ (red curve). **d** Time-dependent absorbance changes of UA ($\lambda = 293$ nm, 0.1 mM) catalyzed by Ni-DAB before and after $SCN^-$ poisoning. **e** O K-edge spectra of Ni-DAB. **f** XPS spectra of C 1s before and after thermal treatment of Ni-DAB at 150 °C in the vacuum. **g** Concentration changes of different electron acceptors in the catalytic oxidation of UA by Ni-DAB. "arb. units" refers to arbitrary units. Source data are provided with the paper.

## Insights of the catalytic mechanism by DFT calculations

DFT calculations were used for a thorough check of all C, N and Ni sites to determine the active center of ORR. Since the *OOH (* representing the chemical adsorbed state) is the first product during ORR, the adsorption energy of *OOH as the indicator to find out the active site was used. *OOH was placed on every site of Ni-DAB and optimized the strutters. The results showed that only the metal center and two beta C sites can have stable *OOH adsorption configurations and the others cannot (Supplementary Fig. 50). The adsorption energies of *OOH displayed in Supplementary Table 5 and the metal center was used as the reference zero. It turned out that the active site for Ni-DAB was the beta C (3) and that for Co-DAB was the Co center. To further understand insights into the UOX-like high catalytic specificity by Ni-DAB through metal-ligand dual sites, density functional theory (DFT) calculations were performed. At first, different typical substrates, including UA, OPD, Glu, 2,4-DP, and ABTS, were explored to investigate the interactions with Ni-DAB in the view of charge transfer (Fig. 4a). It was found that in the cases of OPD, Glu, and 2,4-DP the transferred

charge was negligible, indicating a weak interaction with Ni-DAB at the Ni metal center and subsequently a poor $O_2$ activation at the ligand site. In experiments, no noticeable changes in the EPR spectra for Ni-DAB were observed when the substrates of OPD, Glu, and 2,4-DP were added, indicative of insufficient electron transfer to Ni-DAB (Supplementary Fig. 51 and Note 6). It was on the contrary for UA and ABTS, which had an evident interaction with Ni-DAB as revealed by the DFT calculations. For UA, the binding energy on the Ni metal center was −0.52 eV and an apparent charge transfer from UA to Ni-DAB was observed. This made the beta C in the DAB ligand behave as an active site to reduce $O_2$. While for ABTS, the binding energy was −1.01 eV, resulting the formation of a stable Ni-ABTS complex (see EPR experiments in Supplementary Fig. 52 and Note 6). Such strong adsorption would primarily deactivate the Ni center and impede further reactions. Therefore, the high catalytic specificity of Ni-DAB for UA was derived from not only the delicate interaction between UA and the Ni center but also the complementary oxygen reduction at the beta C site in the ligand.

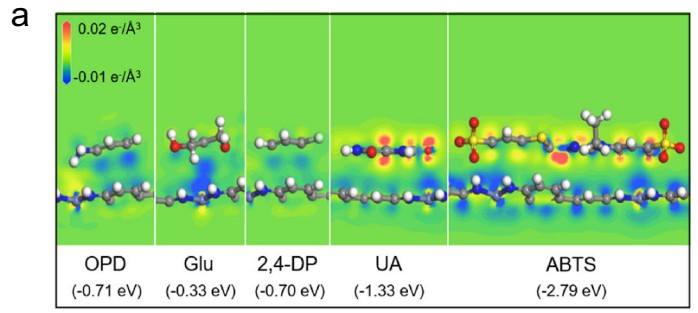

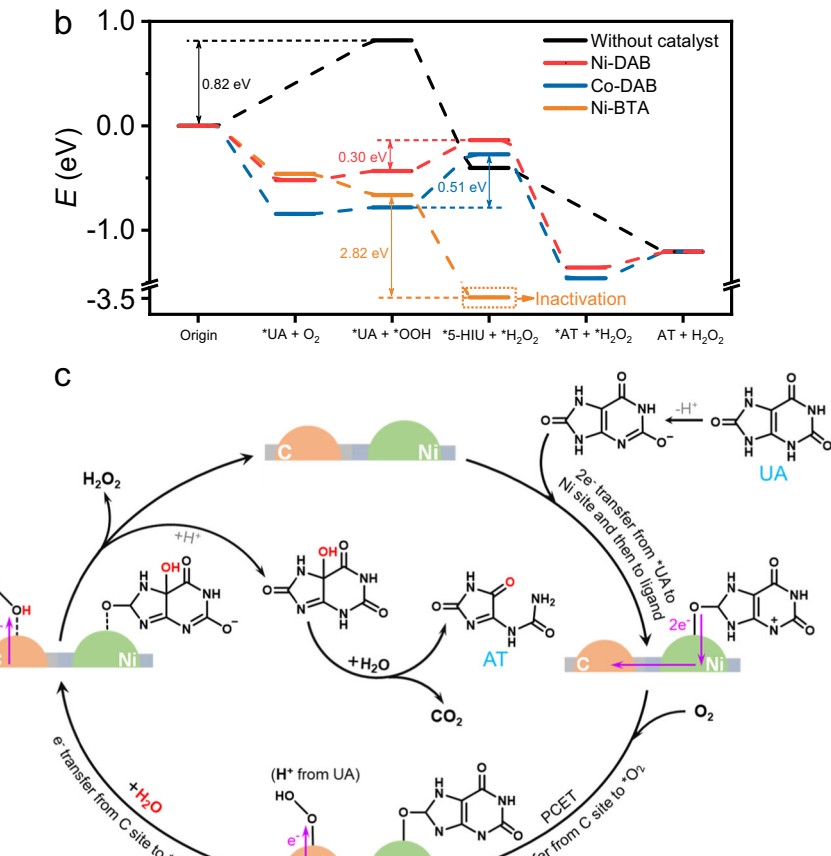

**Fig. 4 | Theoretical calculations and reaction pathways for oxidization of UA by Ni-DAB. a** Charge transfer for different typical substrates adsorption upon Ni-DAB by DFT calculations. The absorption energy is marked under each substrate. The white, grey, blue, red, yellow, and light blue balls represent H, C, N, O, S, and Ni atoms, respectively. **b** Free energy profiles for oxidation of UA catalyzed by Ni-DAB, Co-DAB, Ni-BTA, and without catalyst. **c** Scheme of the proposed reaction pathways for Ni-DAB catalyzed UA oxidation via the metal-ligand dual sites. Source data are provided with the paper.

Figure 4b displays the free energy profiles of the UA oxidation process for the isolated state or on the catalysts of Ni-DAB, Co-DAB, and Ni-BTA. The direct oxidation of UA was difficult on account of the spin-forbidden between the singlet UA and triplet $O_2$, which is demonstrated with a prohibitive energy increase (0.82 eV) in the energy profile. This energy profile was greatly flattened when Ni-DAB got involved. UA docked on Ni-DAB first, and then donated charge to Ni-DAB through the orbital overlap between the 8-O position of UA and Ni center. The transferred charge made Ni-DAB promote a $2e^-$ ORR. $O_2$ could get electrons and be reduced on the beta C site of ligand to form an *OOH intermediate. This *OOH would be reduced into $H_2O_2$ by taking H from an $H_2O$ with ΔE of 0.30 eV. For Co-DAB, the ORR site was the Co center, which has a large endothermic step due to the strong oxygen affinity. Thus, the formation of $H_2O_2$ state was difficult as indicated by the energy profile, and thus the activity would be inferior

to that of Ni-DAB. While for Ni-BTA, the reaction proceeded via different pathways (Fig. 4b). The beta C site of Ni-BTA was very reactive in oxygen reduction, and the formation of *OOH became exothermic by 0.20 eV. Such high activity revealed that the ORR on this site would go through $4e^-$ like route to break the O-O bond of *OOH instead of to form $H_2O_2$. Consequently, a stable carboxyl in the beta C site on Ni-BTA and $H_2O$ were generated by releasing the energy of 2.82 eV. In this sense, rather than UA oxidation, Ni-BTA itself was oxidized and deactivated. We also investigated cases with only Ni sites for UA oxidation. However, these experiments revealed that Ni sites alone cannot catalyze the reaction (Supplementary Fig. 53, 54 and Note 7). These results demonstrated that a reasonable combination of metal centers and ligands was crucial for the dual-site catalytic UA oxidation.

Therefore, the complete UA oxidation pathways catalyzed by Ni-DAB via the metal-ligand dual sites are proposed in Fig. 4c. Briefly, UA

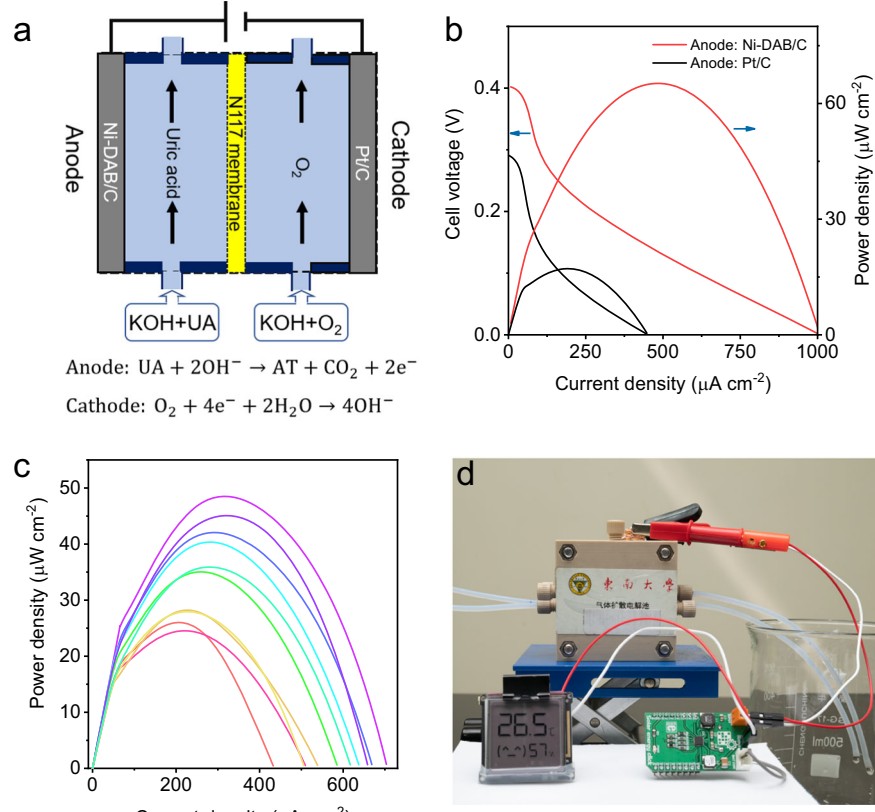

**Fig. 5 | Biofuel cell and metabolite power collection system for sensors.** **a** Schematic diagram of the BFC@Ni-DAB/C device of UOX//ORR. **b** Polarization and power density curves of BFC@Ni-DAB/C and BFC@Pt/C. **c** Power density curves of BFC@Ni-DAB/C using urine as the source of UA. **d** BFC@Ni-DAB/C drives a temperature and humidity sensor. Source data are provided with the paper.

first ionized into a monoanion in aqueous solution, then it was bound to the Ni metal center on Ni-DAB and donated two electrons. Consequently, the electrons transferred from the Ni metal center to the DAB ligand. Afterwards, the proton-coupled electron transfer (PCET) process occurred, i.e., the electron-rich ligand reduced $O_2$ and combined a proton from UA, forming *OOH on the beta C site. The further reduction of *OOH into $H_2O_2$ on the beta C site generated a water-derived hydroxyl group, which attacked the 5-C position of *UA, generating the oxidation product of *5-HIU and then hydrolyzing into AT and $CO_2$. Finally, Ni-DAB returned to its initial state. These processes were similar to that of natural UOX. Notably, if the Ni center deactivates, UA will not bind to it and transfer electrons, and the ligand will not acquire electrons and bind to $O_2$, causing the entire reaction to terminate. Therefore, we successfully mimicked UOX with high catalytic specificity for UA oxidation using Ni-DAB.

**Metabolite power collection system**

Implantable medical devices require a continuous supply of low-power electrical energy and currently use miniaturized batteries, which need to be replaced periodically through surgical procedures. Biofuel cell (BFC) provides a viable solution to continuously supply these devices with electrical energy in vivo. However, BFC using glucose as fuel has the potential risk of competing with cells for food. UA is a metabolic waste product of the body, and BFC using UA as fuel would avoid this risk. Aiming at replacing nature UOX, the as-prepared Ni-DAB SAzymes with high selectivity and catalytic efficiency is suitable for implantable BFC in an anodic reaction, which can reduce side effects and maintain high activity and stability, thereby providing more sufficient electrical energy.

Moreover, converting UA waste into useful compounds and electricity could also find potential applications under extremely conditions, such as in space station and emergency service. Therefore, a UOX//ORR alkaline BFC was assembled, with Ni-DAB/C as the anode and Pt-loaded carbon fiber paper as the cathode (BFC@Ni-DAB/C). As a control, a BFC with Pt/C anode was also assembled (BFC@Pt/C). A diagram of BFC@Ni-DAB/C is shown in Fig. 5a. The power density curves of the BFC are displayed in Fig. 5b. The maximum power output of BFC@Ni-DAB/C reached 65 μW cm$^{-2}$ at 0.14 V, and the open circuit potential was 0.40 V, superior to that of BFC@Pt/C. BFC@Ni-DAB/C also demonstrated good output stability (Supplementary Fig. 55).

The BFC can generate electricity from UA generated in human urine. Urine from several volunteers was collected, the pH was adjusted by adding 0.1 M KOH and then was used as the anode electrolyte for the BFC. The power density curves are shown in Fig. 5c. Using human urine as the source of UA, the power densities of the BFC@Ni-DAB/C were in the range of 25-50 μW cm$^{-2}$, depending on the concentrations of UA in urine samples. To extract usable energy from BFC, an ultra-low-power energy harvester-Solar Energy Click (MIKROE) was used. The click can continuously operate at inputs as low as 100 mV or 5 μW and output a voltage of 3.3 V. As shown in Fig. 5d, BFC@Ni-DAB/C successfully drove a temperature and humidity sensor. Further by interface engineering, it is highly envisioned that such Ni-DAB-based BFC with high stability would work in a physiological environment to offer electricity for implantable medical devices and applicable under emergency service to send a distress radio message. These applications do not involve storing large amounts of urine and have no potential biosafety issues.

## Discussion

In summary, we have developed a metal-ligand dual-site SAzyme (Ni-DAB), which exhibits highly specific catalytic activity for UA oxidation. Comprehensive experiments confirm the Ni metal center and the C atom in the ligand serve as the specific UA and $O_2$ binding sites, respectively. DFT calculations unambiguously suggest the dual-site selection mechanism, in which the Ni center first selectively bound UA to form *UA-Ni-DAB complex, and subsequently $O_2$ readily bound at the beta C site of the DAB ligand for the complete catalytic process. In contrast, other substrates, such as amine, phenolic, catechol, and aldehyde, either failed to bind the Ni center or the beta C site of the formed complex failed to bind oxygen. As a proof-of-the-concept application, a Ni-DAB-based UOX//ORR BFC using human urine was constructed, which successfully drives a temperature and humidity sensor. This work, inspired by the catalytic mechanism of natural UOX, provides a dual-site approach to boost the selectivity of artificial enzymes toward highly efficient and selective catalysis. Notably, loading Ni-DAB onto carbon black (Ni-DAB/C) can achieve better activity and specificity. Other supporters, such as carbon nanotubes/fibers and graphene[77], would offer an additional way to modulate the electronic interactions with metal centers and ligands, which deserves a future investigation.

## Methods

### Synthesis of Ni-DAB

DAB·4HCl (360 mg, 1 mmol) and distilled water (100 mL) were placed in a round-bottomed flask and stirred at room temperature (RT) under air bubbling. A solution of NiCl$_2$·6H$_2$O (238 mg, 1 mmol) in distilled water (50 mL) and concentrated aqueous ammonia (28 wt. %, 3 mL, 45 mmol) was added and stirred for 3 h. After centrifugation, the obtained powder was washed several times with acetone and deionized water and dried under vacuum at 60 °C for 5 h. Elemental analysis calculated for Ni-DAB (wt.%): C, 53.39 %; H, 3.72 %; N, 20.84 %; and Ni, 21.85. Found: C, 46.68 %; H, 3.86 %; Ni, 17.62 %; Ni, 14.81%.

### Synthesis of Ni-DAB/C

DAB·4HCl (360 mg, 1 mmol) and distilled water (100 mL) were placed in a round-bottomed flask and stirred at RT under air bubbling. Ketjenblack EC600J (CB, Akzo Nobel N.V., Netherlands, 240 mg) was then added, and the solution was ultrasonically dispersed. Subsequently, a solution of NiCl$_2$·6H$_2$O (238 mg, 1 mmol) in distilled water (50 mL) and concentrated aqueous ammonia (28 wt. %, 3 mL, 45 mmol) was added and stirred for 3h. After centrifugation, the obtained powder was washed several times with acetone and deionized water and dried under vacuum at 60 °C for 5 h.

### Synthesis of Co-DAB

DAB·4HCl (360 mg, 1 mmol) and distilled water (100 mL) were placed in a round-bottomed flask and stirred at RT under air bubbling. A solution of CoCl$_2$·6H$_2$O (238 mg, 1 mmol) in distilled water (50 mL) and concentrated aqueous ammonia (28 wt. %, 3 mL, 45 mmol) was added and stirred for 3 h. After centrifugation, the obtained powder was washed several times with acetone and deionized water and dried under vacuum at 60 °C for 5 h.

### Synthesis of Ni-BTA

BTA·4HCl (284 mg, 1 mmol) and distilled water (100 mL) were placed in a round-bottom flask and stirred at RT under air bubbling. A solution of NiCl$_2$·6H$_2$O (238 mg, 1 mmol) in distilled water (50 mL) and concentrated aqueous ammonia (28 wt. %, 3 mL, 45 mmol) was added and stirred for 3h. After centrifugation, the obtained powder was washed several times with acetone and deionized water and dried under vacuum at 60 °C for 5 h.

### Kinetic assays for UOX-like activity

The kinetic assays of biomimetic catalysts were monitored by measuring the decrease in the absorbance of UA ($\varepsilon = 13000\ M^{-1}\ cm^{-1}$) at 293 nm using a UV–Vis spectrophotometer in the time-drive mode. The concentration of UA was varied ($10 \times 10^{-6}$, $20 \times 10^{-6}$, $30 \times 10^{-6}$, $40 \times 10^{-6}$, $50 \times 10^{-6}$, $70 \times 10^{-6}$, $150 \times 10^{-6}$, and $200 \times 10^{-6}$ M, dissolved in 3 mM LiOH) at a constant concentration of catalyst (25.0 μg mL$^{-1}$) in NaOH solution (100 mM, pH 13) at 25 °C. The absorbance at 293 nm against the reaction time was plotted to obtain the reaction–time curve. The initial rate of change of absorbance at 293 nm was determined by measuring the slope of the initial linear portion of the reaction–time curve. A Michaelis–Menten curve was produced by plotting the calculated initial rate against the substrate concentrations. The Michaelis–Menten equation was fitted to the data points to determine $V_{max}$ and $K_m$.

### Test of catalytic specificity

The catalytic specificity property of the biomimetic catalysts was tested by some common oxidase substrates using a UV–vis spectrophotometer. The reaction mixture contained the catalyst (50.0 μg mL$^{-1}$) and substrate (0.5 mM) at the most suitable pH for each substrate at 25 °C.

The catalytic specificity property for Glu was tested by the change in pH value. Glu aqueous solution (100 mM) contains catalyst (50.0 μg mL$^{-1}$) and KCl (100 mM). The pH value changes are measured using a pH meter at 25 °C.

### In situ thermally treated NAP-XPS measurement

The NAP-XPS experiments were performed at the NANO-X Vacuum Interconnected Nanotech Workstation. The in situ NAP-XPS measurement was performed using a SPECS NAP-XPS instrument with a temperature-controllable laser heating device and thermocouple equipped. The photon source was the monochromatic X-ray source of Al Kα (1486.6 eV), and the overall spectra resolution was Ag 3d5/2, <0.5 eV FWHM at 20 kcps@UHV. The catalyst powder was pressed onto copper foam, and then the sample was fixed onto the XPS sample holder by tantalum strips. In a routine test, C 1s, O 1s, N 1s, and Ni 2p initial spectra were collected under an ultra-high vacuum at 25 °C, followed by the heat treatment of the sample using the laser at 150 °C in an ultra-high vacuum for 2h, and XPS spectra were collected from the clean surface. Subsequently, the catalyst was exposed to 0.5 mbar $O_2$ in the analysis chamber for 30 min. The collected spectra were fitted using the mixed Gaussian-Lorentzian functions (70% Gaussian and 30% Lorentzian) or Lorentzian Asymmetric Lineshape (C 1s spectrum) after the subtraction of a Shirley background.

### BFC and ultra-low-power energy harvester

The anode for the BFC was prepared as follows: 10 mg Ni-DAB/C and 1 mg PVDF were dispersed into 200 μL of NMP, then 100 μL of the mixture was cast onto a Ni foam (1 cm$^{-2}$). The cathode is Pt loaded carbon fiber paper (0.5 mg cm$^{-2}$, Suzhou Sinero Technology Co., Ltd). To assemble the fuel cell, the anode and cathode were placed in a flow cell separated by a Nafion 117 membrane. The anode electrolyte was UA solution (10 mM) or urine (containing 0.1 M KOH) saturated with $N_2$, and the cathode electrolyte was a 0.1 M KOH solution saturated with $O_2$.

To extract usable energy from BFC, an ultra-low-power energy harvester-Solar Energy Click (MIKROE) was used, which can store energy in a capacitor and increase the output voltage of the BFC. The click is equipped with BQ25570, a nano-power high-efficiency boost charger and buck converter device, designed to work with very low-power energy harvesting elements. The click can continuously operate at inputs as low as 100 mV or 5 μW and output a voltage of 3.3 V. The internal capacitor was used as the energy storage element to continuously power the load. When the high logic level was applied on the OUT pin, the power output for the connected load will be enabled.

## Reporting summary

Further information on research design is available in the Nature Portfolio Reporting Summary linked to this article.

## Data availability

The data supporting the conclusions of this study are present in the paper and the Supplementary Information. The raw data sets used for the presented analysis within the current study are available from the corresponding authors upon request. Source data are provided with this paper.

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

## Acknowledgements

This work was supported by the National Natural Science Foundation of China (22174014 and 22074015). The numerical calculations have been done on the computing facilities in the Nanjing University High Performance Computing Center (NJU HPCC). We are grateful for the technical support from Dr. Yifan Li from Nano-X of Suzhou Institute of Nano-Tech and Nano-Bionics, Chinese Academy of Sciences, for NAP-XPS measurements.

## Author contributions

Y.Z., and K.W. conceived and designed the experiments. K.W. performed the synthesis, characterization, activity evaluation, mechanism studies, and application of Ni-DAB and Ni-DAB/C SAzymes. Q.H., C.Z. and Y.X. participated in the activity evaluation. W.L., Y.W., Y.F., W.C. and X.G. assisted in preparation of SAzymes. All authors contributed to the analysis and discussion of the results. K.W., and Y.Z. co-wrote the manuscript, and K.W., Q.H., C.Z., X.C., Y.X., S.L., Y.S., and Y.Z. revised the manuscript. All authors reviewed the manuscript. Y.Z. supervised the project.

## Competing interests

The authors declare no competing interests.
