## [Peer Review File · Nature Communications]

Metal-ligand dual-site single-atom nanozyme mimicking urate oxidase with high substrates specificityREVIEWER COMMENTS

Reviewer #1 (Remarks to the Author):

This study reveals that the Ni-DAB complex formed by the coordination of Ni and DAB exhibits specific catalytic activity in uric acid oxidation. Compared to the commercial Pt/C catalyst, Ni-DAB demonstrates higher efficiency and specificity in catalyzing uric acid oxidation. The authors suggest that Ni and C atoms in Ni-DAB act as catalytic sites for binding uric acid and O₂, respectively, following a catalytic mechanism like uricase. Based on the catalytic capabilities of Ni-DAB, the authors have developed a metabolite power collection system with potential applications in the development of implantable medical devices. I acknowledge the potential value of this research in the development of urease mimetics. However, some key information about the configuration determination of Ni-DAB and DFT theoretical calculations is not clearly expounded in the text. I therefore suggest the following major revision:

1. In the Introduction section, replace "single-atom enzymes" with "single-atom nanozyme."
2. In the Introduction section, the statement "non-protein artificial enzymes lack any recognition sites for specific substrates" is inappropriate as the results of this study and some others contradict this claim.
3. In the Introduction section, the statement "such catalysts (coordination polymers) have been rarely reported" is misleading, as many catalysts based on coordination polymers have been reported.
4. In the Results section, the text "The UOX is critical for the metabolism of UA, which allows some bacteria to use UA as the only carbon and nitrogen source and is closely related to the nitrogen fixation reaction of nitrogen-fixing plants. Humans and primates have lost the ability to express UOX because of evolutionary accidents and are vulnerable to hyperuricemia." seems unrelated to the subsequent research results. Consider moving it to the Introduction section or removing it.
5. Is there any electrostatic adsorption between the Ni-DAB, which is positively charged, and the uric acid, which is negatively charged? Were potential interferences caused by substrate electrostatic adsorption excluded during the measurement of catalytic activity?
6. Theoretically, the configuration with two Ni-coordinated DAB molecules in a plane perpendicular to each other is the lowest energy configuration compared to those depicted

in Figs. 1a,b and S10. The authors are recommended to calculate the most stable molecular configurations of Ni-DAB using kinetic simulation. In other words, the authors should first determine whether the Ni-N₄ coordination in the Ni-DAB molecule is planar or steric before analyzing the catalytic mechanism.

7. Is the Ni in Ni-DAB coordinated saturated? Did the authors consider other coordination except N coordination for making the Ni coordinate saturated when performing DFT theoretical calculations? In the SACs system, Cl can usually regulate the catalytic performance through axial coordination. However, in this work, with NiCl₂ as the precursor and without pyrolysis treatment, Cl will not be removed, which can be confirmed by element-mapping. According to the literature, this method of direct adsorption without pyrolysis suggests that Cl may adsorb onto the metal center in an axial adsorption manner (nano research, 2022, 15, 10063-10069). Therefore, the authors need to explain what structure Cl exists in? And what effect does the presence of Cl have on the catalytic activity?

8. When measuring the catalytic kinetics, did the authors consider the potential influence of O₂ concentration in the measurement system on catalytic kinetics during the measurement of catalytic dynamics?

9. The statement "The specific activity value of Ni-DAB was determined to be 1.51 U mg⁻¹, which was significantly higher than those controls of Co-DAB, Ni-BTA, and previously reported artificial UOX mimics (Supplementary Table. 3)" is inappropriate because Supplementary Table. 3 lists kinetic parameters, not specific activity.

10. Does the structure and coordination of Ni-DAB change with uricase substrates or products? Due to the absence of high-temperature treatment in the preparation of this catalyst, it may not be very stable during the reaction process. As shown in Figure S17, the activity significantly decreases with the reduction of pH. Therefore, the authors need to supplement the structure and catalytic stability after the reaction.

11. Suggest adding a schematic diagram to Figure S28 to illustrate the experimental purpose.

12. The purpose and mechanism of adding carbon black in the synthesis should be written in the text.

13. How did the authors determine that beta-C but not other carbon is the active center for binding O₂?

14. In Figure S29 and Note S3, consider changing "DMPO-•O₂-" to "DMPO-OOH" for clarity.

15. In the non-pyrolyzed Ni-DAB catalyst, Ni center acts as a metal node within a stable catalyst structure, as opposed to being embedded in the carbon substrate lattice. It is essential to highlight that the pyrolysis process typically results in the contraction of Ni-N bonds. For instance, in pyrolysis-prepared Ni-N-C SACs commonly cited in literature, bond lengths have been observed to decrease from 2.09 Å to 1.86 Å (J. Am. Chem. Soc. 2017, 139, 42, 14889–14892). Remarkably, the catalyst achieved through the polymerization process alone exhibits a Ni-N bond length of 1.84Å. To validate the structural accuracy of these findings, it is recommended that authors disclose the parameters of their theoretical model, including bond lengths and angles, and conduct a comparison between these modeled bond lengths and those determined through XAFS fitting.

16. I believe it is inappropriate to poison with KSCN in alkaline 0.1M KOH, because for the Ni²⁺ center, a small amount of SCN⁻ competes with a large amount of OH⁻ for adsorption. Therefore, the unchanged ORR activity observed by the authors may not necessarily be the result of SCN⁻ poisoning. I suggest the authors conduct the poisoning experiment under acidic conditions, such as in H₂SO₄ electrolyte.

Reviewer #2 (Remarks to the Author):

Zhang and coworkers have designed and synthesized a metal-ligand compound that mimics the dual-site catalytic activities of urate oxidase. They have shown that the metal and carbon sites of the compound serve as the specific binding sites for UA and O₂, respectively, and the catalysis has remarkably selectivity for uric acid over other substrates OPD, Glu, 2,4-DP, and ABTS. The mechanism of the catalysis and selectivity has been well studied by experiments as well as DFT calculations. The manuscript is well written and is recommended for publication at Nat Commun after the authors consider the following points.

1. In Fig 4a, the charge distributions for different substrates are shown to explain the selective adsorption of UA to Ni-DAB. It is suggested to mark the adsorption energy for each substrate in the same figure.
2. The definitions of 5-HIU (5-Hydroxyisourate) and AT seem to be missing.
3. Ni-DAB has high selectivity for UA over OPD, Glu, 2,4-DP, and ABTS. How about the selectivity between UA and phenol compounds, as deprotonated phenol anions may also

have good affinity for the Ni site of the catalyst?

Reviewer #3 (Remarks to the Author):

I have checked the entire manuscript on S-zyme and related revealing of active sites by using soft X-ray in situ and XPS techniques. I do not find this work as novel as a paper in Nat. Commun. deserves due to following reasons:

- 1) BFC with human urine and active sites of Ni-DAB has unclear links between them which does not contribute to novelty of work.
- 2) Why authors chose a Ni-DAB system which is a molecular Ni-atom system whereas general single Ni atom systems are much more interesting in terms of electronics.
- 3) Non-protein S-zyme can be also be made with Ni-SA on CNF or graphene and electronic tuning can be achieved. In this respect, what Ni-DAB is offering which remain unclear.
- 4) Beta C sites adsorbs O-species due to what reasons which remain unclear.
- 5) Deactivation scope of Ni site is not substantiated by experiment.

Point-by-point response to the comments for Manuscript NCOMMS-24-10530-T

For Reviewer #1:

Q: This study reveals that the Ni-DAB complex formed by the coordination of Ni and DAB exhibits specific catalytic activity in uric acid oxidation. Compared to the commercial Pt/C catalyst, Ni-DAB demonstrates higher efficiency and specificity in catalyzing uric acid oxidation. The authors suggest that Ni and C atoms in Ni-DAB act as catalytic sites for binding uric acid and O₂, respectively, following a catalytic mechanism like uricase. Based on the catalytic capabilities of Ni-DAB, the authors have developed a metabolite power collection system with potential applications in the development of implantable medical devices. I acknowledge the potential value of this research in the development of urease mimetics. However, some key information about the configuration determination of Ni-DAB and DFT theoretical calculations is not clearly expounded in the text. I therefore suggest the following major revision:

A: We thank this reviewer for offering us valuable suggestions, which we could make a substantial improvement to this manuscript. The detailed response to the comments is shown as follows.

Q1: In the Introduction section, replace "single-atom enzymes" with "single-atom nanozyme."

A1: Sorry for the confusion. In the revised manuscript, we have replaced "single-atom enzymes" with "*single-atom nanozymes*."

Q2: In the Introduction section, the statement "non-protein artificial enzymes lack any recognition sites for specific substrates" is inappropriate as the results of this study and some others contradict this claim.

A2: For clarity, "non-protein artificial enzymes lack..." has been revised as "*most non-protein artificial enzymes lack...*" (Page 3).

Q3: In the Introduction section, the statement "such catalysts (coordination polymers) have been rarely reported" is misleading, as many catalysts based on coordination polymers have been reported.

A3: Thank you for your suggestion. It has been revised as "*Many catalysts based on coordination polymers have been reported; nonetheless, the studies on enhancing the specificity of enzyme-like reaction based on polymeric catalysts are rare.*" (Page 4).

Q4: In the Results section, the text "The UOX is critical for the metabolism of UA, which allows some bacteria to use UA as the only carbon and nitrogen source and is closely related to the nitrogen fixation reaction of nitrogen-fixing plants. Humans and primates have lost the ability to express UOX because of evolutionary accidents and are vulnerable to hyperuricemia." seems unrelated to the subsequent research results. Consider moving it to the Introduction section or removing it.

A4: By following this suggestion, such redundant text has been removed.

Q5: Is there any electrostatic adsorption between the Ni-DAB, which is positively charged, and the uric acid, which is negatively charged? Were potential interferences caused by substrate electrostatic adsorption excluded during the measurement of catalytic activity?

A5: Good question! For this, we additionally tested the zeta potential of Ni-DAB after adding substrates with positive or negative charges. Interestingly, the dependence of zeta potentials on the surface charge of substrates was found, indicating there was an electrostatic interaction between Ni-DAB and substrates with different charges. However, considering only UA was selectively oxidized, here the electrostatic interaction was not the origin of specificity.

Therefore, for a better scholarly presentation, "*The addition of negatively charged substrates (UA, ABTS, etc) can cause zeta potential changes of Ni-DAB (Supplementary Fig. 40), but only UA can be catalyzed, indicating the selectivity was not caused by electrostatic adsorption.*" has been added in the revised manuscript (Page

12), and in the revised Supplementary Information, **Supplementary Fig. 40** has been added.

Supplementary Fig. 40. Zeta potential changes of Ni-DAB with different substrates added. [New data]

Q6: Theoretically, the configuration with two Ni-coordinated DAB molecules in a plane perpendicular to each other is the lowest energy configuration compared to those depicted in Figs. 1a,b and S10. The authors are recommended to calculate the most stable molecular configurations of Ni-DAB using kinetic simulation. In other words, the authors should first determine whether the Ni-N4 coordination in the Ni-DAB molecule is planar or steric before analyzing the catalytic mechanism.

A6: Very valuable question! Based on the valence bond theory, when coordinating with DAB, the Ni^{2+} with d^8 electronic configuration adopts dsp^2 hybridization to form an inner orbital complex with a square-planar coordination configuration. For instance, Ni^{2+} in square-planar coordination configurations were reported in several previous works, e.g., *Dincă et al., J. Am. Chem. Soc. 2019, 141, 10475* and *Peng et al., Angew. Chem. Int. Ed. 2020, 59, 286*. Therefore, the square-planar coordination should be reasonable for Ni-DAB.

To verify this assumption, we have performed a first principle molecular dynamics (MD) simulation to make sure the stable configuration of Ni-DAB is properly obtained. A piece of Ni-DAB chain with two DAB molecules was used as the model. After 500 steps of MD at 300K, the configuration of Ni-DAB largely remains the initial configuration. The final frame is extracted and optimized, which gives the same

configuration as the initial one. We also built a long chain Ni-DAB and optimized it with the DFT method, which also gives a similar dihedral angle to the models used in this manuscript. By the way, our model also has similar configurations to the ones reported in the literature.

Therefore, for a better scholarly presentation, “*The stable configuration of Ni-DAB was further verified by the first principle of molecular dynamics, consistent with EXAFS fitting (Supplementary Fig 12 and Note 3).*” has been added in the revised manuscript (Page 7). In the revised Supplementary Information, **Supplementary Note 3** and **Supplementary Fig. 12** have been added.

Supplementary Note 3. *The first principle of molecular dynamics (MD) was used to imitate the stable configuration of Ni-DAB. A piece of Ni-DAB chain with two DAB molecules was used as the model. After 500 steps of MD at 300K, the configuration of Ni-DAB largely remained the initial configuration. The final frame was extracted and optimized, which gave the same configuration as the initial one (Supplementary Fig. 12). The long chain Ni-DAB was also built and optimized with the DFT method, which also showed a similar dihedral angle to the models. By the way, the model also has similar configurations to the ones reported in the literature^{5,6}.*

Supplementary Fig. 12. *Stable configuration of Ni-DAB imitated by the first principle of molecular dynamics (MD). After 500 steps of MD at 300K, the final frame was extracted and optimized, which showed the same configuration as the initial one. **a**, Before MD. **b**, After MD. The white, grey, blue, and light blue balls represent H, C, N, and Ni atoms, respectively. [New data]*

Q7: Is the Ni in Ni-DAB coordinated saturated? Did the authors consider other coordination except N coordination for making the Ni coordinate saturated when performing DFT theoretical calculations? In the SACs system, Cl can usually regulate the catalytic performance through axial coordination. However, in this work, with NiCl₂ as the precursor and without pyrolysis treatment, Cl will not be removed, which can be confirmed by element-mapping. According to the literature, this method of direct adsorption without pyrolysis suggests that Cl may adsorb onto the metal center in an axial adsorption manner (nano research, 2022, 15, 10063-10069). Therefore, the authors need to explain what structure Cl exists in? And what effect does the presence of Cl have on the catalytic activity?

A7: Very valuable suggestion. Following this advice, firstly, we supplemented a DFT modeling with one or two Cl atoms coordinated with Ni of Ni-DAB. It turned out that the equilibrium Ni-Cl bond lengths (2.530 Å for single Cl atom absorption, 2.767 Å and/ 2.760 Å for two Cl atoms absorption) were larger than 2.5 Å. In this case, the Ni center was generally not coordinated with Cl, and they interact through ionic interactions.

Secondly, we additionally measured the amount of Cl element before and after catalytic oxidation of UA by Ni-DAB. It was found that Cl⁻ in Ni-DAB was almost completely replaced by OH⁻ after catalysis, indicating Cl⁻ only acted as an equilibrium charger in the freshly as-prepared Ni-DAB, rather than coordinating with the metal center of Ni-DAB in an axial adsorption manner. In the text “Synthesis and structural characterizations of Ni-DAB” section, the characterization and theoretical modeling have been detailed.

Therefore, for a better scholarly presentation, “*Furthermore, after UA was catalyzed by Ni-DAB in alkaline conditions, Cl⁻ in Ni-DAB was almost completely replaced by OH⁻, suggesting that Cl⁻ only had an equilibrium charge effect, did not coordinate to the metal center of Ni-DAB in an axial adsorption manner⁶⁵ (Supplementary Fig. 9, Note 2 and Table 1), and had no effect on the catalytic performance.*” has been added

into the revised manuscript, along with the citation of the references as Ref. 65. (Page 6).

Accordingly, **Supplementary Fig. 9**, **Table 1**, and **Note 2** have been added to the revised Supplementary Information.

Supplementary Fig. 9. Structures of single (a) and two (b) Cl ion(s) adsorbed onto the metal center of Ni-DAB in an axial adsorption manner. The distances between Ni and Cl were too far to be coordinated. The white, grey, blue, green, and light blue balls represent H, C, N, Cl, and Ni atoms, respectively. [New data]

Supplementary Table 1. XPS atomic ratio of Ni-DAB before and after catalytic UA oxidation.

Atomic ratio	C%+N%+Ni%	O%	Cl%
Ni-DAB (before)	92.23	3.93	3.84
Ni-DAB (after)	92.10	7.47	0.43
Variation value		+3.54	-3.41

Supplementary Note 2. The Cl element has been reported to serve as an axial ligand for Ni SACs and regulate catalytic activity⁵. Therefore, the models of Cl in the axial position of Ni were calculated. After structural optimization, it was found that the distances between Ni and Cl were too far (2.530 Å for single Cl atom absorption, 2.767/2.760 Å for two Cl atoms absorption) to be coordinated (**Supplementary Fig. 9**). [New discussion]

Q8: When measuring the catalytic kinetics, did the authors consider the potential influence of O₂ concentration in the measurement system on catalytic kinetics during the measurement of catalytic dynamics?

A8: Very valuable suggestion. Following this advice, we have tested catalytic kinetics under different O₂ concentrations. As shown in **Supplementary Fig. 16**, Ni-DAB simulates oxidase, whose activity depends on the oxygen concentration in the system.

Therefore, we have added “*Ni-DAB catalyzed UA oxidation with dependence on oxygen concentration like natural oxidase. In the air-saturated solution, the calculated value for V_{max} was $0.68 \mu\text{M s}^{-1}$.*” in the revised manuscript (Page 9).

In the revised Supplementary Information, we have added **Supplementary Fig. 16**

Supplementary Fig. 16. Michaelis–Menten curves for catalytic UA oxidation by Ni-DAB. The solution was bubbled with N₂ for 5 min and the concentration of dissolved O₂ was 3.4 mg/L. Ni-DAB catalyzes UA oxidation with dependence on oxygen concentration like natural oxidase, and the catalytic rate slows down when the oxygen content in the solution decreases. **[New data]**

Q9: The statement "The specific activity value of Ni-DAB was determined to be 1.51 U mg⁻¹, which was significantly higher than those controls of Co-DAB, Ni-BTA, and previously reported artificial UOX mimics (Supplementary Table. 3)" is inappropriate because Supplementary Table. 3 lists kinetic parameters, not specific activity.

A9: Sorry for the poor presentation and confusion. For clarity, it has been revised as “Ni-DAB catalyzed UA oxidation with dependence on oxygen concentration like natural oxidase. In the air-saturated solution, the calculated value for V_{max} was $0.68 \mu\text{M s}^{-1}$. The UOX-mimic activity of Ni-DAB was significantly higher than those controls of Co-DAB, Ni-BTA (**Supplementary Fig. 17**), and previously reported artificial UOX mimics (**Supplementary Table. 4**). The specific activity value of Ni-DAB was determined to be 1.51 U mg^{-1} (**Supplementary Fig. 18**).” in the revised manuscript (Page 9-10)

Q10: Does the structure and coordination of Ni-DAB change with uricase substrates or products? Due to the absence of high-temperature treatment in the preparation of this catalyst, it may not be very stable during the reaction process. As shown in Figure S17, the activity significantly decreases with the reduction of pH. Therefore, the authors need to supplement the structure and catalytic stability after the reaction.

A10: Very valuable suggestion. According to this advice, we measured the catalytic recyclability and structure stability of Ni-DAB in the catalytic oxidation of UA. It was found the UOX-like activity of Ni-DAB was stable and recyclable (**Supplementary Fig. 22**). Moreover, as shown in **Supplementary Fig. 23**, XPS spectra confirmed that the molecular structure of Ni-DAB did not alter during the catalysis UA oxidation.

Therefore, for a better scholarly presentation, “The UOX-like activity of Ni-DAB was stable and recyclable (**Supplementary Fig. 22**). The XPS analysis disclosed that the Ni 2p, C 1s and N 1s spectra of the spent Ni-DAB show no obvious changes compared with those of the fresh Ni-DAB (**Supplementary Fig. 23**). These results reconfirmed its robust structural stability during the catalytic UA oxidation.” have been added in the revised manuscript (Page 10). Accordingly, In the revised Supplementary Information, **Supplementary Fig. 22** and **23** have been added.

Supplementary Fig. 22. Recovery and recyclability of Ni-DAB in the catalytic oxidation of UA. Error bars represented the standard error derived from three independent measurements. [New data]

Supplementary Fig. 23. High-resolution XPS spectra of Ni-DAB before and after repetitive catalytic oxidation of UA. **a**, Ni 2p. **b**, C 1s. **c**, N 1s. [New data]

Q11: Suggest adding a schematic diagram to Figure S28 to illustrate the experimental purpose.

A11: Thank you for your suggestion. According to this advice, we have added a diagram to Figure S34 (Previous S28) to illustrate the experimental purpose.

Supplementary Fig. 34. UV-vis absorption spectra of solutions with combinations of TMB, HRP, and the supernatant reaction solution (RS, diluted 40 times) for the preparation of Ni-DAB. Inset: diagram of the by-product H_2O_2 produced during the synthesis of Ni-DAB. [Updated data]

Q12: The purpose and mechanism of adding carbon black in the synthesis should be written in the text.

A12: Thank you for your suggestion. In the revised manuscript, Supplementary Note 3, i.e., “In **Supplementary Fig. 34**, when HRP and TMB were added to the supernatant reaction solution (RS) for the synthesis of Ni-DAB. An obvious characteristic adsorption peak of oxidized TMB (TMB_{ox}) appeared at 652 nm, implying that H_2O_2 was generated in the solution. This explained why the Ni-DAB ligand was in an electron-deficient oxidation state, because electrons from the ligand were transferred to O_2 during the synthesis of Ni-DAB. Furthermore, trace amounts of H_2O_2 were detected in the aqueous dispersion supernatant (SN) of as-synthesized Ni-DAB. And, the amount of H_2O_2 increased under O_2 -saturated conditions, while decreased under N_2 -saturated conditions (**Supplementary Fig. 35a**). EPR trapping measurements were further performed to detect free oxygen radicals in solution containing Ni-DAB. **Supplementary Fig. 35b** showed a signal typical of the DMPO-OOH adduct was detected in Ni-DAB dispersion. These results confirmed the generation of reactive

oxygen species (ROS) from O_2 reduction by the as-synthesized Ni-DAB. We speculated that not all ligands in Ni-DAB were oxidized during synthesis and some of them continued to reduce O_2 in as-synthesized Ni-DAB, which generated ROS and resulted in an imperfect catalytic specificity in substrates. Therefore, to further improve the catalytic specificity, carbon black was added during the synthesis of Ni-DAB (Ni-DAB/C, **Supplementary Fig. 36 and 37**), which can increase the contact area between the DAB ligands and O_2 during the preparation process, making the Ni-DAB ligand oxidation be more complete.” has moved to the main text in the revised manuscript (Page 11), to highlight the purpose and mechanism of adding carbon black in the synthesis.

Q13: How did the authors determine that beta-C but not other carbon is the active center for binding O_2 ?

A13: Very valuable suggestion. Based on the experiment data (**Fig. 3b-f and Supplementary Fig. 42-48**), we have determined that the ORR site of Ni-DAB is the non-metallic C site. To further verify it in theory, the DFT calculations were carried out. It has been reported that beta-C is most likely to serve as the ORR site in Ni- N_4 coordination polymers (Ref. 63: *Dinca et al., ACS Catal. 2017, 7, 7726*). In our initial simplified model, the chemical environment of the two beta-C (2, 3) in Ni-DAB was consistent, thus we chose the beta-C (2) site with lower steric hindrance as the ORR site for calculation.

Supplementary Fig. 50. Diagram of possible ORR active sites of Ni-DAB, including Ni metal center (1), beta-C (2), and beta-C (3) [**New scheme**]

Taking inspiration from the reviewer's suggestion, we did a thorough check of all C, N and Ni sites to determine the active center. Since the *OOH was the first product

during ORR, we then used the adsorption energy of *OOH as the indicator to find out the active site. We placed *OOH on every site of Ni-DAB and optimized the structures. The results showed that the metal center and two beta-C sites can have stable *OOH adsorption configurations and the others cannot. The adsorption energies of *OOH are displayed in **Supplementary Table 5** and the metal center is used as the reference zero. It turned out that the active site for Ni-DAB was the beta-C (3) and that for Co-DAB was the Co metal center.

The new results can well explain the different catalytic behavior of Ni-DAB and Co-DAB. For Ni-DAB, the beta-C (3) had a decent energy profile, while that for Co-DAB had a large endothermic step due to the strong oxygen affinity of the Co center.

Therefore, for a better scholarly presentation, we have made the following revisions:

- (1) *“DFT calculations were used for a thorough check of all C, N and Ni sites to determine the active center of ORR. Since the *OOH (* representing the adsorbed state) is the first product during ORR, the adsorption energy of *OOH as the indicator to find out the active site was used. *OOH was placed on every site of Ni-DAB and optimized the structures. The results showed that only the metal center and two beta-C sites can have stable *OOH adsorption configurations and the others cannot (Supplementary Fig. 50). The adsorption energies of *OOH displayed in Supplementary Table 5 and the metal center was used as the reference zero. It turned out that the active site for Ni-DAB was the beta-C (3) and that for Co-DAB was the Co center.”* has been added in the revised manuscript (**Pages 15-16**).
- (2) In the revised manuscript, **Fig. 1a,b** has been updated.
- (3) In the revised manuscript, **Fig. 4b** has been updated.

(4) “This *OOH would be spontaneously reduced into H_2O_2 by taking H from an H_2O with ΔE of -0.11 eV. For Co-DAB, the formation of *OOH state was difficult as indicated by the energy profile, and thus the activity would be inferior to that of Ni-DAB.” has been modified as “*This *OOH would be reduced into H_2O_2 by taking H from an H_2O with ΔE of 0.30 eV. For Co-DAB, the ORR site was the Co center, which has a large endothermic step due to the strong oxygen affinity. Thus, the formation of H_2O_2 state was difficult as indicated by the energy profile, and thus the activity would be inferior to that of Ni-DAB.*” (Page 19).

(5) The discussion of “According to the free energy, this was the rate-determining step” has been removed (Page 19).

(6) In the revised Supplementary Information, **Supplementary Fig. 50** has been added.

Supplementary Fig. 50. Diagram of possible ORR active sites of Ni-DAB, including Ni metal center (1), beta-C (2), and beta-C (3)

(7) In the revised Supplementary Information, **Supplementary Table 5** has been added.

Supplementary Table 5. Free energy of stable adsorption of *OOH at different sites

Entry	Site 1: Ni (eV)	Site 2: beta-C (eV)	Site 3: beta-C (eV)
Ni-DAB	0	0.27	-0.18
Co-DAB	0	0.86	0.66

The ball-stick structure is shown in Supplementary Fig. 50. Free energy of the metal center is used as the reference zero. [New data]

Q14: In Figure S29 and Note S3, consider changing "DMPO-•O2-" to "DMPO-OOH" for clarity.

A14: Thank you for your suggestion. In the revised Supplementary Information, we have replaced "DMPO-•O₂⁻" with "DMPO-OOH".

Q15: In the non-pyrolyzed Ni-DAB catalyst, Ni center acts as a metal node within a stable catalyst structure, as opposed to being embedded in the carbon substrate lattice. It is essential to highlight that the pyrolysis process typically results in the contraction of Ni-N bonds. For instance, in pyrolysis-prepared Ni-N-C SACs commonly cited in literature, bond lengths have been observed to decrease from 2.09 Å to 1.86 Å (J. Am. Chem. Soc. 2017, 139, 42, 14889–14892). Remarkably, the catalyst achieved through the polymerization process alone exhibits a Ni-N bond length of 1.84Å. To validate the structural accuracy of these findings, it is recommended that authors disclose the parameters of their theoretical model, including bond lengths and angles, and conduct a comparison between these modeled bond lengths and those determined through XAFS fitting.

A15: Very valuable suggestion. According to this advice, we have carefully read the paper (Ref. 67: J. Am. Chem. Soc. 2017, 139, 42, 14889–14892). The paper mentions that the Ni-N bond of the Ni-doped g-C₃N₄ precursor was 2.09 Å, which was due to the

doublet feature of the Ni-N bond, involving two kinds of chemical bonds. The fitting of EXAFS revealed the Ni in the precursor coordinated by two O atoms at 1.95 Å and four N atoms at 2.09 Å. In contrast, the pyrolysis-prepared Ni-N₄-C only showed a dominant peak. The fitted result indicated Ni coordinated by four N atoms at 1.86 Å. In this sense, the pyrolysis process preserved the nitrogen chemical bond and removed the oxygen bond.

By following the advice, we have re-examined our XAS data and theoretical modeling fitting. As shown in Fig. 1g and h, the Ni K-edge EXAFS of Ni-DAB presents a unimodal in the first shell like the pyrolysis-prepared Ni-N₄-C. This indicated that the Ni center coordinates with a single element, and the Ni-N bond demonstrated a short length like that in the pyrolysis-prepared Ni-N₄-C. A similar result was also reported by other groups (Ref. 60: *Peng et al., Angew. Chem. Int. Ed.* 2020, 59, 286).

Therefore, for clarity, the following revision has been made:

- (1) The parameters of the theoretical model of Ni-DAB have been added in the revised **Supplementary Fig. 11**.

Supplementary Fig. 11. Ni-DAB fitting model. One Ni metal center coordinated with multiple nitrogen atoms (ca. 1.84 Å) in the first shell within one plane. [New data]

- (2) In the revised manuscript “one Ni metal center coordinated with multiple nitrogen atoms in the first shell and multiple benzene ring carbon atoms in the second shell within one plane” has been removed to the discussion for **Supplementary Fig. 11**”.
- (3) “Importantly, unlike M-N-C catalysts, Ni-DAB utilized a pyrolysis-free strategy with a well-defined coordination structure.” has been modified as “*The chelation of amino groups made Ni in Ni-DAB present a unimodal in the first shell like the*

pyrolysis-prepared Ni-N-C SACs⁶⁷, enabling Ni-DAB to only have the Ni-N coordination.” in the revised manuscript (Page 7), along with the citation of the above two references as Ref. 60 and 67.

Q16: I believe it is inappropriate to poison with KSCN in alkaline 0.1M KOH, because for the Ni²⁺ center, a small amount of SCN⁻ competes with a large amount of OH⁻ for adsorption. Therefore, the unchanged ORR activity observed by the authors may not necessarily be the result of SCN⁻ poisoning. I suggest the authors conduct the poisoning experiment under acidic conditions, such as in H₂SO₄ electrolyte.

A16: Very valuable suggestion. However, unfortunately, Ni-DAB was instable under acidic conditions, thus for a more comprehensive presentation, we have tested the poisoning experiment of ORR in neutral pH conditions. As shown in **Supplementary Fig. 44b**, in a neutral pH electrolyte, SCN⁻ did not impair the ORR performance of Ni-DAB/C.

In addition, we found some previous reports (Ref. 74: *Proc. Natl. Acad. Sci. USA* 2023, 120, e2308035120; *Small* 2023, 19, 2300289) also utilized SCN⁻ to poison Ni metal site in 0.1 M KOH electrolyte. Therefore, for a better scholarly presentation, we have modified the sentence “SCN⁻ did not impair the ORR performance of Ni-DAB/C (**Supplementary Fig. 44**)” to “SCN⁻ did not impair the ORR performance of Ni-DAB/C in both alkaline and neutral conditions (**Supplementary Fig. 44**)” in the revised manuscript (page 14). In the revised Supplementary Information, we have updated the **Supplementary Fig. 44**.

Supplementary Fig. 44. ORR activity of Ni-DAB/C before and after SCN^- poisoning examined by LSV curves using RDE. **a**, in alkaline 0.1 M KOH. **b**, in neutral 0.05 M PBS. [Updated data]

For Reviewer #2:

Zhang and coworkers have designed and synthesized a metal-ligand compound that mimics the dual-site catalytic activities of urate oxidase. They have shown that the metal and carbon sites of the compound serve as the specific binding sites for UA and O_2 , respectively, and the catalysis has remarkably selectivity for uric acid over other substrates OPD, Glu, 2,4-DP, and ABTS. The mechanism of the catalysis and selectivity has been well studied by experiments as well as DFT calculations. The manuscript is well written and is recommended for publication at Nat Commun after the authors consider the following points.

A: We thank this reviewer for offering us valuable suggestions, which we could make a substantial improvement to this manuscript. The detailed response to the comments is shown as follows.

Q1: In Fig 4a, the charge distributions for different substrates are shown to explain the selective adsorption of UA to Ni-DAB. It is suggested to mark the adsorption energy for each substrate in the same figure.

A1: Thank you for your suggestion. According to your advice, we have updated **Fig. 4a** in the revised manuscript:

Accordingly, in the caption of **Fig. 4a**, “*The absorption energy is marked under each substrate.*” has been added.

Q2: The definitions of 5-HIU (5-Hydroxyisourate) and AT seem to be missing.

A2: Sorry for the poor presentation. By following the suggestion, we have updated the definitions “*5-hydroxyisourate (5-HIU)*” (Page 13) and “*allantoin (AT)*” (Page 9) in the revised manuscript.

Q3: Ni-DAB has high selectivity for UA over OPD, Glu, 2,4-DP, and ABTS. How about the selectivity between UA and phenol compounds, as deprotonated phenol anions may also have good affinity for the Ni site of the catalyst?

A3: Very valuable suggestion. According to your suggestion, we have investigated the catalytic activity and selectivity between UA and typical phenol compounds. As shown in **Fig. 2e** and **Supplementary Fig. 28, 31-33**, the catalytic activity of Ni-DAB towards typical phenol compounds and catechol compounds, such as 2,4-dichlorophenol (2,4-DP), epinephrine (E), dopamine (DA), and 3,5-ditert-butyl catechol (DTBC) was negligible.

Taking 2,4-DP as an example, the DFT calculation (**Fig. 4a**) and the EPR spectra (**Supplementary Fig. 51**) showed that few charges were transferred from 2,4-DP to the Ni site, indicating a weak interaction between 2,4-DP and the Ni site and subsequently a poor O₂ activation at the ligand site.

Therefore, for a better scholarly presentation, “*such as amine, phenolic, catechol, and aldehyde*” has been added in the revised manuscript (Page 22).

For Reviewer #3:

I have checked the entire manuscript on S-zyme and related revealing of active sites by using soft X-ray in situ and XPS techniques. I do not find this work as novel as a paper in Nat. Commun. deserves due to following reasons:

A: We thank the reviewer for the valuable suggestions, which we could make a substantial improvement to this manuscript. The detailed response to the comments is shown as follows.

Q1: BFC with human urine and active sites of Ni-DAB has unclear links between them which does not contribute to novelty of work.

A1: Thank you for your suggestion. As known, natural enzymes suffer from being costly to store, unstable, and sensitive to harsh physiochemical conditions. Thus, single-atom nanozymes (SAzymes), as potential substitutes for natural enzymes, are attracting increasing interest. In this manuscript, we reported that Ni-DAB UOX-mimic has efficient activity and selectivity. Comprehensive experiments and DFT calculations revealed an unusual mechanism of metal-N_x/ligand dual active sites in activating two substrates (i.e., UA and O₂), rather than commonly reported single metal-N_x active site. It well explained why high selectivity of UA oxidation was exceptionally achieved for SAzymes in this work. Compared to previous SAzymes, this unique feature is more suitable to replace natural UOX in an anodic reaction for implantable BFC, because it can reduce side effects and accordingly, improve the output of electrical energy. As a proof-of-the-concept application, a biofuel cell using human urine was successfully constructed.

For a better scholarly presentation, therefore, more discussion of “*Aiming at replacing nature UOX, the as-prepared Ni-DAB SAzymes with high selectivity and catalytic efficiency is suitable for implantable BFC in an anodic reaction, which can reduce side effects and maintain high activity and stability, thereby providing more sufficient electrical energy.*” in the revised manuscript (Page 20).

Q2: Why authors chose a Ni-DAB system which is a molecular Ni-atom system whereas general single Ni atom systems are much more interesting in terms of electronics.

A2: Thank you for your suggestion. Single-atom catalysts (SACs) as emerging high-performance catalysts were widely studied. The traditional SACs require high-temperature pyrolysis, which provides them with excellent performance (e.g., high catalytic activity and single atom economy), but also makes structural analysis difficult, especially in non-metallic regions. This is not conducive to the interpretation of structure-activity relationships and the rational design of catalysts.

To address the problem of poor specificity of the SAzymes, we propose introducing a competitive non-metallic site into SACs, but this is a forbidden task for the pyrolysis-prepared SACs. Old polymeric catalysts (Ref. 58: *Storck et al., Angew. Chem. Int. Ed. 1978, 17, 657*) that have precise engineerable molecular structures come to our attention. Among them, the coordination polymers (CPs) with uniform distribution, high density, unambiguous single-atom framework of metal centers and ligands would be a class of ideal candidates for studying the non-metallic site of SACs. As one example, we chose a molecular Ni-atom system (Ni-DAB). As revealed by the XAFS fitting (**Fig. 1h**), the chelation of amino groups made Ni in Ni-DAB present a unimodal in the first shell like the pyrolysis-prepared Ni-N-C SACs. This is the reason that we chose the molecular Ni-DAB system instead of pyrolyzed single-atom Ni catalysts.

By following the suggestion, for a better scholarly presentation, “Since non-metallic active sites are commonly existed in natural enzymes, one viable biomimicking strategy is to introduce a competitive non-metallic site into SACs^{56, 57}. However, traditional metal-nitrogen-doped carbon (M-N-C) SACs are prepared by high temperature pyrolysis with randomly dispersed metal atoms and no clear continuous local structures. This prevents them from being precisely engineerable. In this case, coordination polymers (CPs) with uniform distribution, high density, unambiguous single-atom framework of metal centers and ligands, and ligand-centered redox activity would be a class of ideal candidates⁵⁸⁻⁶³; however, such catalysts have been rarely reported.” has

been revised as “*Since non-metallic active sites are commonly existed in natural enzymes, one viable biomimicking strategy is to introduce a competitive non-metallic site into SACs^{56, 57}, and the primary prerequisite is to choose a catalyst with an unambiguous structure. However, traditional metal-nitrogen-doped carbon (M-N-C) SACs are prepared by high-temperature pyrolysis with randomly dispersed metal atoms and no clear continuous local structures. This prevents them from being precisely engineerable. Old polymeric catalysts that have precise engineerable molecular structures come to our attention⁵⁸. Among them, coordination polymers (CPs) with uniform distribution, high density, unambiguous single-atom framework of metal centers and ligands, and ligand-centered redox activity would be a class of ideal candidates for studying the non-metallic site of SACs⁵⁹⁻⁶⁴. Many catalysts based on coordination polymers have been reported; nonetheless, the studies on enhancing the specificity of enzyme-like reaction based on polymeric catalysts are rare.*” in the revised manuscript, along with the citation of the above reference as Ref. 58 (Page 4).

Q3: Non-protein S-zyme can be also be made with Ni-SA on CNF or graphene and electronic tuning can be achieved. In this respect, what Ni-DAB is offering which remain unclear.

A3: Thank you for your suggestion. In **A2**, we have explained why we chose the Ni-DAB system in this work. We also loaded Ni-DAB onto carbon black (CB, Ni-DAB/C), which was aimed at obtaining better activity and selectivity, owing to larger specific surface area and more complete ligand oxidation (see **Fig. 2e** and **Supplementary Fig. 38, 39** and discussion in the text of **Page 11**). For clarity, we chose Ni-DAB without CB substrate in mechanism studies. This situation is like Pt electrode for fundamental mechanism studies and Pt loaded onto CB (Pt/C) for practical applications (*Wei et al., Chem. Soc. Rev., 2015, 44, 2168*). As you pointed out, different substrates, such as CNF and graphene, would offer an additional way to modulate the electronic interactions with metal centers and ligands (e.g., Ref. 78: *Wang et al., Nat. Synthesis, 2023, 2, 1194*), which deserves a future investigation.

Therefore, for a better scholarly presentation, “Notably, loading Ni-DAB onto carbon black (Ni-DAB/C) can achieve better activity and specificity. Other supporters, such as carbon nanotubes/fibers and graphene⁷⁸, would offer an additional way to modulate the electronic interactions with metal centers and ligands, which deserves a future investigation.” has been added in the revised manuscript, along with the citation of the above reference as Ref. 78. (Page 22).

Q4: Beta C sites adsorbs O-species due to what reasons which remain unclear.

A4: Very valuable suggestion. Based on the experiment data (**Fig. 3b-f and Supplementary Fig. 42-48**), we have determined that the ORR site of Ni-DAB is the non-metallic C site. To further verify it in theory, the DFT calculations were carried out. It has been reported that beta-C is most likely to serve as the ORR site in Ni-N₄ coordination polymers (Ref. 63: *Dinca et al., ACS Catal. 2017, 7, 7726*). In our initial simplified model, the chemical environment of the two beta-C (2, 3) in Ni-DAB was consistent, thus we chose the beta-C (2) site with lower steric hindrance as the ORR site for calculation.

Supplementary Fig. 50. Diagram of possible ORR active sites of Ni-DAB, including Ni metal center (1), beta-C (2), and beta-C (3) [New scheme]

Taking inspiration from the reviewer's suggestion, we did a thorough check of all C, N and Ni sites to determine the active center. Since the *OOH was the first product during ORR, we then used the adsorption energy of *OOH as the indicator to find out the active site. We placed *OOH on every site of Ni-DAB and optimized the structures. The results showed that the metal center and two beta-C sites can have stable *OOH adsorption configurations and the others cannot. The adsorption energies of *OOH are displayed in **Supplementary Table 5** and the metal center is used as the reference zero.

It turned out that the active site for Ni-DAB was the beta-C (3) and that for Co-DAB was the Co metal center.

The new results can well explain the different catalytic behavior of Ni-DAB and Co-DAB. For Ni-DAB, the beta-C (3) had a decent energy profile, while that for Co-DAB had a large endothermic step due to the strong oxygen affinity of the Co center.

Therefore, for a better scholarly presentation, we have made the following revisions:

(1) “DFT calculations were used for a thorough check of all C, N and Ni sites to determine the active center of ORR. Since the *OOH (* representing the adsorbed state) is the first product during ORR, the adsorption energy of *OOH as the indicator to find out the active site was used. *OOH was placed on every site of Ni-DAB and optimized the strutters. The results showed that only the metal center and two beta-C sites can have stable *OOH adsorption configurations and the others cannot (**Supplementary Fig. 50**). The adsorption energies of *OOH displayed in **Supplementary Table 5** and the metal center was used as the reference zero. It turned out that the active site for Ni-DAB was the beta-C (3) and that for Co-DAB was the Co center.” has been added in the revised manuscript (**Pages 15-16**).

(2) In the revised manuscript, **Fig. 1a,b** has been updated.

(3) In the revised manuscript, **Fig. 4b** has been updated.

(4) “This *OOH would be spontaneously reduced into H₂O₂ by taking H from an H₂O with ΔE of -0.11 eV. For Co-DAB, the formation of *OOH state was difficult as indicated by the energy profile, and thus the activity would be inferior to that

of Ni-DAB.” has been modified as “*This *OOH would be reduced into H₂O₂ by taking H from an H₂O with ΔE of 0.30 eV. For Co-DAB, the ORR site was the Co center, which has a large endothermic step due to the strong oxygen affinity. Thus, the formation of H₂O₂ state was difficult as indicated by the energy profile, and thus the activity would be inferior to that of Ni-DAB.*” (Page 19).

(5) The discussion of “According to the free energy, this was the rate-determining step” has been removed (Page 19).

(6) In the revised Supplementary Information, **Supplementary Fig. 50** has been added.

Supplementary Fig. 50. Diagram of possible ORR active sites of Ni-DAB, including Ni metal center (1), beta-C (2), and beta-C (3)

(7) In the revised Supplementary Information, **Supplementary Table 5** has been added.

Supplementary Table 5. Free energy of stable adsorption of *OOH at different sites

Entry	Site 1: Ni (eV)	Site 2: beta-C (eV)	Site 3: beta-C (eV)
Ni-DAB	0	0.27	-0.18
Co-DAB	0	0.86	0.66

The ball-stick structure is shown in Supplementary Fig. 50. Free energy of the metal center is used as the reference zero. **[New data]**

Q5: Deactivation scope of Ni site is not substantiated by experiment.

A5: Sorry for the poor presentation. We have explored the UOX-like activity of Ni-DAB under different temperatures and pH conditions (**Supplementary Fig. 21**). It was found that Ni-DAB remained activity over a broad pH value (6 ~ 14) and temperature (10 ~ 80 °C) range. When the pH value was below 6, Ni-DAB became unstable and inactive. Control experiments using natural UOX and UA showed that SCN^- had no reaction activities on substrates and products (**Supplementary Fig. 42**). In this sense, the inhibited catalytic oxidation activity of Ni-DAB (**Fig. 4d**) was from the SCN^- poisoning effect at Ni site.

To further verify the above assumption, we have added an ORR poison experiment at a neutral pH electrolyte. As shown in **Supplementary Fig. 44**, the SCN^- did not impair the ORR active centers of Ni-DAB under both neutral and alkaline conditions. Therefore, for a better scholarly presentation, we made the following revisions:

- (1) In the revised manuscript, “when the pH value was below 6, Ni-DAB became unstable and inactive” has been added (Page 10), and “The control experiment showed that SCN^- had...” has been revised as “Control experiments using natural UOX and UA showed that SCN^- had...” (Page 13).
- (2) In the revised Supplementary Information, we have updated **Supplementary Fig. 44**.

Supplementary Fig. 44. ORR activity of Ni-DAB/C before and after SCN^- poisoning examined by LSV curves using RDE. **a**, in alkaline 0.1 M KOH. **b**, in neutral 0.05 M PBS. [Updated data]

REVIEWER COMMENTS

Reviewer #1 (Remarks to the Author):

The authors have fully addressed my concerns.

Reviewer #2 (Remarks to the Author):

All my concerns have been well addressed.

Reviewer #3 (Remarks to the Author):

I have the following comments from the revised manuscript by the authors:

- 1) It will be more clear if a model can be compared which offers only metal or ligand site instead of dual sites as done by the DFT analysis. How does the free-energy profile will change in such single-site case.
- 2) Storing human urine in real form for such BFC would raise a question mark on affecting the environment in lab or manufacturing unit. How would you work on the biosafety of such process.
- 3) Deactivation of Ni center will impede what kind of further reactions.

Point-by-point responses to the comments for Manuscript NCOMMS-24-10530A

Reviewer #3 (Remarks to the Author):

Q: I have the following comments from the revised manuscript by the authors:

A: We thank this reviewer for offering us valuable suggestions, which we could make a substantial improvement to this manuscript. The detailed response to the comments is shown as follows.

Q1: It will be more clear if a model can be compared which offers only metal or ligand site in stead of dual sites as done by the DFT analysis. How does the free-energy profile will change in such single-site case.

A1: Very valuable suggestion.

As control, we supplemented the calculation that UA oxidation and ORR catalyzed by the same Ni site (Ni single-site) on Ni-DAB. Interestingly, the energy profile shows that this process is quite energetically favorable (**Supplementary Fig. 53**). For comparison, we also performed a calculation that UA oxidation and ORR catalyzed by the different Ni sites (Ni-Ni dual-site) on Ni-DAB. This time, there is an energy increase of 0.35 eV for the step of $O_2 \rightarrow *OOH$, indicating the poor to activate O_2 . These results suggest that the Ni center with the axial ligand can facilitate the ORR on it. However, we think such a single metal site catalyzed process is not dominant in the practical situation. Due to the high affinity of the Ni center for UA, the Ni center will be occupied by two UA molecules on either side of Ni-DAB (-0.53 eV and -0.55 eV) and O_2 cannot access the Ni center due to steric hindrance (**Supplementary Fig. 54**). In contrast, such UA occupation has no steric effect on the beta C site of Ni-DAB, which can provide reliable ORR site during reactions.

For Co-DAB, DFT analysis in the main text showed that Co-DAB has only the metal active site for both the UA binding and ORR. As shown in **Fig. 4b**, the Co metal site has a large endothermic step in $*OOH \rightarrow H_2O_2$ due to the strong oxygen affinity. Thus,

the formation of the H₂O₂ state was difficult as indicated by the energy profile (endothermic 0.51 eV), and thus the activity would be inferior to that of Ni-DAB.

Therefore, for a better scholarly presentation:

- (1) *“We also investigated cases with only Ni sites for UA oxidation. However, these experiments revealed that Ni sites alone cannot catalyze the reaction (Supplementary Fig. 53, 54 and Note 7).”* has been added in the revised manuscript (page 19), and in the revised Supplementary Information, **Supplementary Fig. 53, 54 and Note 7** has been added.

- (2) *Supplementary Note 7, i.e., “As controls, the calculation that UA oxidation and ORR catalyzed by the same Ni site (Ni single-site) on Ni-DAB was supplemented. Interestingly, the energy profile showed that this process is quite energetically favorable (Supplementary Fig. 53). For comparison, the calculation that UA oxidation and ORR catalyzed by the different Ni sites (Ni-Ni dual-site) on Ni-DAB was also performed. This time, there is an energy increase of 0.35 eV for the step of O₂→*OOH, indicating the poor to activate O₂. These results suggested that the Ni center with the axial ligand can facilitate the ORR on it. However, we think such a single metal site catalyzed process is not dominant in the practical situation. Due to the high affinity of the Ni center for UA, the Ni center will be occupied by two UA molecules on either side of Ni-DAB (-0.53 eV and -0.55 eV) and O₂ cannot access the Ni center due to steric hindrance (Supplementary Fig. 54). In contrast, such UA occupation has no steric effect on the beta C site of Ni-DAB, which can provide reliable ORR site during reactions.”* has been added into the revised Supplementary Information.

(3) Supplementary Fig. 53

Supplementary Fig. 53. a, Free energy profiles for UA oxidation catalyzed by Ni-DAB with Ni-C dual-site, Ni-Ni dual-site, and Ni single-site. **b**, Diagram for the Ni single-site of Ni-DAB, in which UA and O₂ bind to the same Ni center. **c**, Diagram for the Ni-Ni dual-site of Ni-DAB, in which UA and O₂ bind to adjacent two Ni centers respectively. The white, grey, blue, red, and light blue balls represent H, C, N, O, and Ni atoms, respectively.

(4) Supplementary Fig. 54

Supplementary Fig. 54. Free energy profile for the adsorption of UA on the same Ni center. The inset shows the structure of two UA adsorbed onto the same Ni center of Ni-DAB. The white, grey, blue, red, and light blue balls represent H, C, N, O, and Ni atoms, respectively.

Q2: Storing human urine in real form for such BFC would raise a question mark on affecting the environment in lab or manufacturing unit. How would you work on the biosafety of such process.

A2: Very valuable suggestion. According to our vision, such BFC would offer electricity for implantable medical devices in a physiological environment and provide electrical energy under extremely conditions, such as in space station and emergency service, these applications do not involve the storage of large amounts of urine and have no potential biosafety issues.

Therefore, for a better scholarly presentation, "...and applicable under emergency service to send a distress radio message." has been revised as "...and applicable under emergency service to send a distress radio message. These applications do not involve storing large amounts of urine and have no potential biosafety issues." in the manuscript (page 21).

Q3: Deactivation of Ni center will impede what kind of further reactions.

A3: Very valuable suggestion. Due to the high specificity of Ni-DAB for UA, the deactivated Ni center cannot bind to UA and transfer electrons, then the ligand DAB cannot obtain electrons. Thus, the beta-C site of ligand cannot bind to O₂, and the entire reaction would stop.

Therefore, for a better scholarly presentation, “Finally, Ni-DAB returned to its initial state. These processes were similar to that of natural UOX; therefore, we successfully mimicked UOX with high catalytic specificity for UA oxidation using Ni-DAB.” has been revised as “*Finally, Ni-DAB returned to its initial state. These processes were similar to that of natural UOX. Notably, If the Ni center deactivates, UA will not bind to it and transfer electrons, and the ligand will not acquire electrons and bind to O₂, causing the entire reaction to terminate. Therefore, we successfully mimicked UOX with high catalytic specificity for UA oxidation using Ni-DAB.*” in the manuscript (page 20).

REVIEWERS' COMMENTS

Reviewer #3 (Remarks to the Author):

The revision required was performed as per the comments. I am ok with the acceptance of this ms for publication.